# GIRAFE v1: a global climate data record for precipitation accompanied by a daily sampling uncertainty

Hannes Konrad[1], Rémy Roca[2], Anja Niedorf[1], Stephan Finkensieper[1], Marc Schröder[1], Sophie Cloché[3], Giulia Panegrossi[4], Paolo Sanò[4], Christopher Kidd[5,6], Rômulo Augusto Jucá Oliveira[2,*], Karsten Fennig[1], Thomas Sikorski[1], Madeleine Lemoine[2], Rainer Hollmann[1]

[1]Deutscher Wetterdienst, Satellite-Based Climate Monitoring, Frankfurter Str. 135, 63067 Offenbach am Main, Germany
[2]Laboratoire d'Etudes Géophysiques et d'Océanographie Spatiales, 14, av. Edouard Belin, 31401 Toulouse cedex 9, France
[3]Institut Pierre-Simon Laplace, Sciences du Climat, route de Saclay, 91128 Palaiseau, France
[4]National Research Council of Italy, Institute of Atmospheric Sciences and Climate, Via del Fosso del Cavaliere 100, 00133 Roma, Italy
[5]University of Maryland, Earth System Science Interdisciplinary Center, 5825 University Research Ct., College Park, MD College Park, MD, USA
[6]NASA/Goddard Space Flight Center, 8800 Greenbelt Rd, Greenbelt, MD 20771, USA
[*]Now at: Hydro Matters, 1 Chemin de la Pousaraque, 31460 Le Faget, France

*Correspondence to*: Hannes Konrad (hannes.konrad@dwd.de)

**Abstract.** Here we introduce the first version of the Global Interpolated RAinfall Estimation (GIRAFE v1), the first dedicated global climate data record for precipitation by the Satellite Application Facility on Climate Monitoring (CM SAF) of the European Organisation for the Exploitation of Meteorological Satellites (EUMETSAT). GIRAFE is based on precipitation rate estimates obtained from observations by a variety of passive microwave radiometers (PMW) onboard low-Earth orbit satellites and related retrieval algorithms and frequent and highly resolved infrared observations from geostationary satellites covering all longitudes and used at latitudes below 55°N/S. At higher latitudes, only the PMW-based precipitation rates are utilized. GIRAFE v1 is available globally at 1-degree resolution as daily accumulations and monthly means for the years 2002-2022, with an implementation of continuous production planned for 2025 onwards. The daily product is accompanied by a dedicated sampling uncertainty estimate based on decorrelation scales in space and time in infrared-based instantaneous precipitation fields. The methods for the generation of GIRAFE v1 are described in detail, followed by results of quality assessment and intercomparison activities. GIRAFE v1 reproduces reference datasets with a performance similar to established precipitation products, especially as those that are – like GIRAFE v1 – not adjusted to ground-based observations. Likewise, GIRAFE v1 proves to be suitable for the analysis of regional precipitation extremes, e.g. in their relation to sea surface temperatures. The main objective in the production of GIRAFE v1 is climate applications, for which we find the dataset highly suitable according to the stability and homogeneity analysis. The GIRAFE v1 data record is hosted by CM SAF and is freely available at https://doi.org/10.5676/EUM_SAF_CM/GIRAFE/V001 (Niedorf et al., 2024a).

## 1 Introduction

Precipitation estimates from space have emerged as a critical tool for both academic research and operational applications, providing unique insights into the Earth's hydrological cycle. From an academic point of view, space-based observations allow the study of precipitation patterns on a global scale, which is essential for understanding climate dynamics, weather systems, and water resource management. Satellites equipped with advanced sensors, such as active and passive microwave instruments, consistently provide comprehensive data, enabling the analysis of precipitation processes across different scales – ranging from local storms to the global climate. Furthermore, space-based precipitation monitoring aids in refining the quantification of the global water budgets and supports long-term climate studies that help assess the impact of anthropogenic climate change (Stephens et al. 2022). From an operational standpoint, satellite precipitation estimates complement ground-based radar and rain-gauge networks used by operational agencies, especially in regions where ground observations are sparse or unavailable. This global coverage is crucial for monitoring of extreme weather events, such as floods and droughts, which have significant societal and economic impact (Levizzani et al. 2018).

Over the last two decades, monitoring precipitation from space has benefited from the emergence of a constellation of satellites with microwave observation capabilities, a sustained and improved fleet of operational meteorological geostationary satellites, and mature retrieval algorithms (Levizzani and Cattani 2019). There is now a large number of precipitation datasets spanning that period or longer (Roca et al. 2019). While climate monitoring has usually been performed at monthly time scales, monitoring precipitation distributions at daily scales is important for extreme events. Over land, satellite-based precipitation products incorporating rain-gauge data perform better overall than reanalysis-based products (Bador et al. 2020; Alexander et al. 2020). Recent assessments nevertheless point to the need for better products in view of the academic and societal challenges ahead (Roca et al. 2021).

Here, we introduce the result of a coordinated effort at the European scale around EUMETSAT's Satellite Application Facility for Climate Monitoring (CM SAF), elaborating and distributing a new satellite precipitation climate data record (CDR), the Global Interpolated RAinfall Estimaton version 1 (GIRAFE v1). Our product is generated by an algorithm applied in an operational environment to multi-source observations from the historical record (2002-2022) as well as the recent past (monthly updates with a three-month latency, becoming available in 2025). This paper aims at introducing this new product and its performance with emphasis on climate compliancy metrics.

Section 2 explains the rationale of the production. Section 3 introduces the data feeding into GIRAFE v1, while the methodology is detailed in section 4. Section 5 is dedicated to the evaluation of the product against various reference and non-reference datasets using climate-focused metrics. Section 6 describes data access for GIRAFE v1 and respective input data as

well as for the various datasets which are used and evaluated together with GIRAFE v1 in section 5. Finally, section 7 provides

a conclusion.

## 2 Data generation overview

The overall data processing (Figure 1) is built around the GIRAFE algorithm. The GIRAFE algorithm combines instantaneous

infrared brightness temperatures (Tb) (Level-1) observed by geostationary satellites (section 3.1) and instantaneous

precipitation rate estimates (Level-2) provided by passive microwave (PMW) radiometers onboard Low Earth Orbit (LEO)

satellites (section 3.2). Both input streams undergo dedicated pre-processing: quality control of the instantaneous infrared Tbs

(section 4.1) and a homogenisation of the instantaneous precipitation rate estimates via quantile mapping (section 4.2).

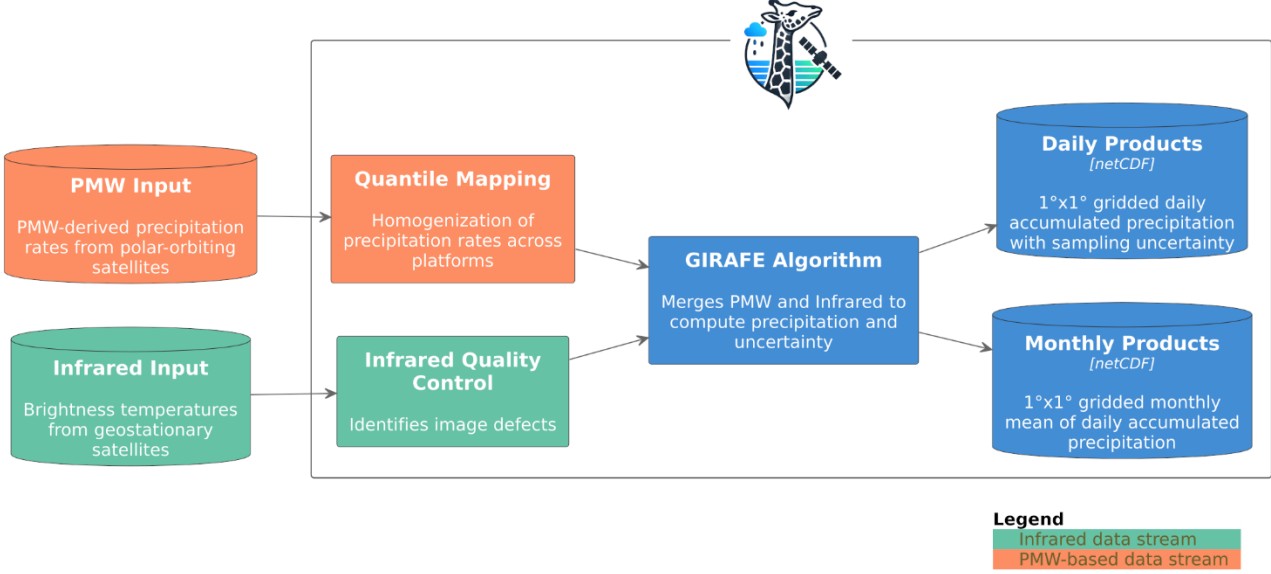

**Figure 1: Overview of the algorithm and data streams.**

For latitudes inside 55°N/S, the GIRAFE algorithm merges the two input data streams towards the gridded daily accumulated

precipitation output (section 4.3.1). Grid cells outside of 55°N/S are based only on the PMW-based data stream (section 4.3.2).

The limit for using the infrared data stream is defined at 55°N/S because the distortion of the fields of view becomes larger

towards the poles due to increasing viewing angles, leading to a trade-off between the improved temporal sampling and the

declining accuracy of the infrared observations. This is similar to the latitudinal boundary used in GPCP v3.2 (Huffman et al.,

2023a). The uncertainty module of the GIRAFE algorithm uses the variance in the daily precipitation fields and intermediate

results of the merging module for a quantification of the number of independent samples to derive the sampling uncertainty

associated with the daily accumulated precipitation product (section 4.4). The monthly mean daily accumulated precipitation
is derived directly from the daily fields (section 4.5).

## 3 Input data

### 3.1 Geostationary Level 1 infrared brightness temperatures

Passive infrared imagers onboard at least five geostationary satellites have been providing a quasi-global coverage with the above-mentioned latitudinal restrictions. While there have been gaps in the constellation of geostationary satellites prior to
1998, it has been complete in the time period considered for GIRAFE v1 (2002 onwards). All infrared imagers feature an infrared channel between 10.3 and 11.5 microns with spatial resolutions between 2 and 5 km at nadir, with temporal samplings between 10 and 30 minutes, including partial scans of the Northern and Southern Hemispheres. Figure 2 provides an overview of the available platforms. Instrument details are summarized in Table 1.

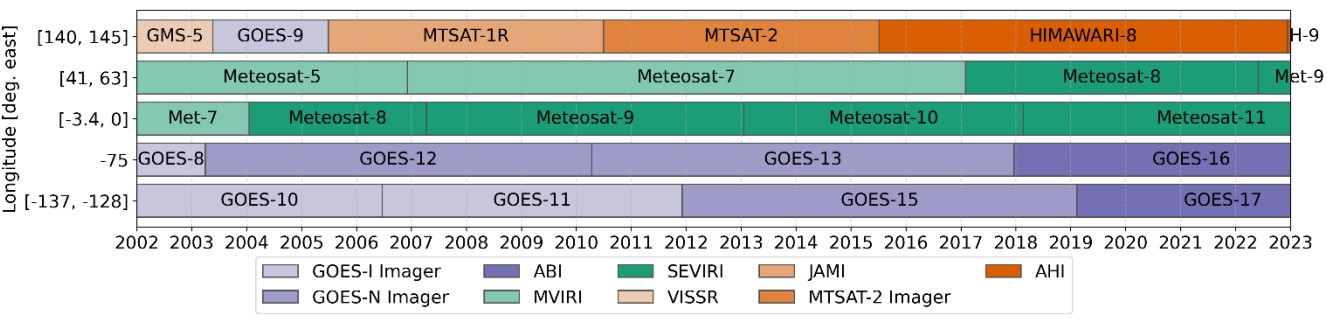


**Figure 2: Geostationary platforms and their temporal coverage as used in GIRAFE v1. The respective instruments are color-coded, see also Table 1. The labels on the vertical axis indicate the sub-satellite longitude range of the platforms in the respective position. H-9 = HIMAWARI-9, Met-9 = Meteosat-9.**

Instantaneous infrared Tb observations (Level-1) for the entire timeseries have been collected from the corresponding agencies
(EUMETSAT, NOAA, JMA/JAXA). In principle, despite GIRAFE being designed for climate applications, the infrared Tb observations are not required from high-quality intercalibrated Fundamental CDR (FCDR) archives: long-term drifts in the infrared observations are intercepted by the training of the infrared observations with PMW-derived precipitation rate estimates over short windows (one-day periods, see section 4.3.1). Therefore, non-intercalibrated data from operational archives may be used. Operational calibration coefficients are applied to convert raw image counts to Tbs. As one available exception,
EUMETSAT's FIDUCEO MVIRI FCDR (Rüthrich et al. 2020a,b,c) for Meteosat First Generation is preferred over the operational data, due to the improved (inter-)calibration and the accompanying quality control. Where possible, future GIRAFE versions will rely on more infrared datasets of FCDR quality.

**Table 1: Geostationary instruments used in GIRAFE v1. [†] Reduced temporal resolution to limit computation time.**

| Platform Series | Instrument (infrared imager) | Channel (μm) | Spatial resolution at nadir (km) | Temporal resolution (min) |
|---|---|---|---|---|
| GMS-5 | Visible/Infrared Spin Scan Radiometer (VISSR) | 11.0 | 5 | 60 |
| MTSAT-1R | Japanese Advanced Meteorological Imager (JAMI) | 10.8 | 4 | 30 |
| MTSAT-2 | MTSAT-2 Imager | 10.8 | 4 | 30 |
| HIMAWARI | Advanced Himawari Imager (AHI) | 10.4 | 2 | 20[†] |
| Meteosat First Generation | Meteosat Visible Infra-Red Imager (MVIRI) | 11.5 | 5 | 30 |
| Meteosat Second Generation | Spinning Enhanced Visible and Infra-Red Imager (SEVIRI) | 10.8 | 3 | 15 |
| GOES-I/N | GOES-I/N Imager | 10.7 | 4 | 30 |
| GOES-R | Advanced Baseline Imager (ABI) | 10.3 | 2 | 15/20[†] |

## 3.2 Passive microwave-based Level 2 precipitation rate estimates

The PMW-derived database used for GIRAFE v1 is diverse in the sense that we use the data obtained from nine different types of PMW radiometers operated on a series of 19 different LEO satellites, processed by three different algorithms. The diversity in their retrievals, such as the frequency distribution of the occurrence of precipitation rates makes the pre-processing described in section 4.2 necessary. Figure 3 lists all PMW satellites that feed into GIRAFE v1. Table 2 provides details on the instruments. The respective retrieval algorithms and data sources are detailed in sections 3.2.1-3.2.3.

**3.2.1 HOAPS**

Instantaneous precipitation rates from PMW imagers over ice-free ocean are based on the Hamburg Ocean Atmosphere Parameters and fluxes from Satellite (HOAPS) dataset. For the precipitation rate retrieval, an artificial neural network (ANN) trained with precipitation rates retrieved from assimilated Tbs in a 1D-Var scheme from the European Centre for Medium Range Weather Forecast (ECMWF) has been designed to derive a statistical retrieval from SSM/I and SSMIS (SSMI(S) from

here) Tbs (Andersson et al., 2010). The resulting HOAPS precipitation retrieval is a statistical algorithm that only depends on Tbs as input data. As argued by Andersson et al. (2010), precipitation rates below 0.3 mm/h are set to zero due to their low signal-to-noise ratio, i.e. where the microwave signatures possibly stemming from (wind-driven) surface emission modulations or the presence of cloud liquid water or water vapour are misinterpreted as those of precipitation or vice versa (e.g., Ferraro et

al., 1998). The ANN is designed for PMW imager Tbs of the CM SAF SSMI(S) FCDR (Fennig et al., 2020) at the resolution
of the 37 GHz channel of the respective sensor.

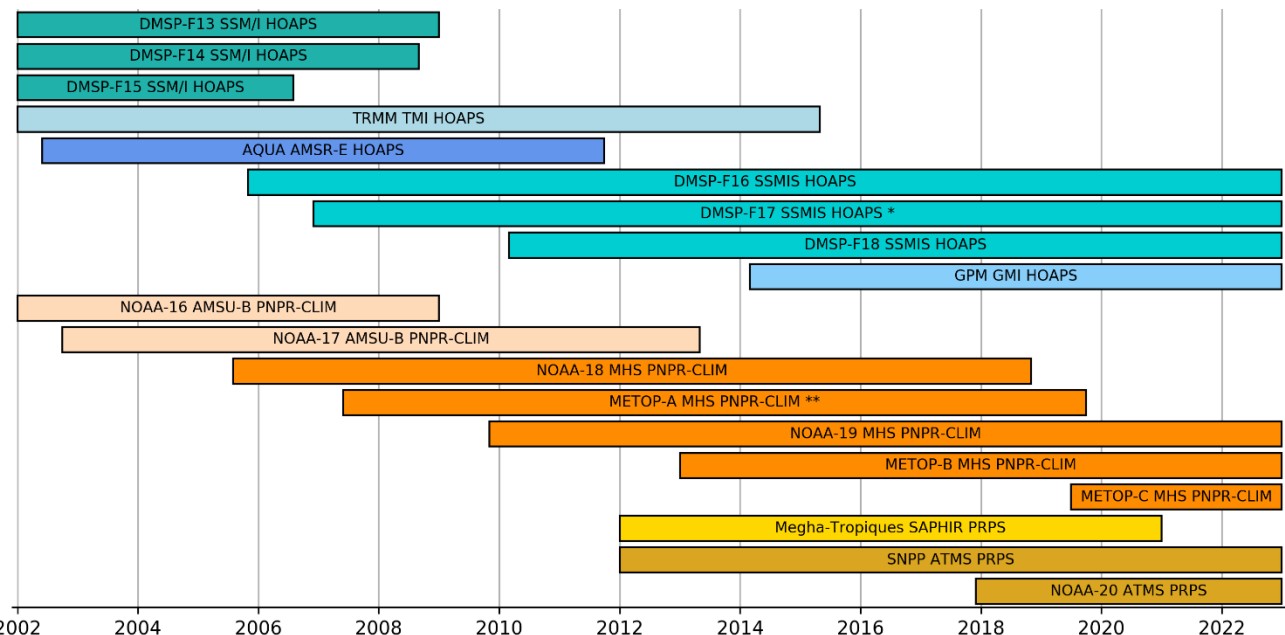

**Figure 3: Overview over satellites bearing PMW sensors and their temporal availability used in GIRAFE v1. The labels indicate satellite, instrument and the corresponding precipitation retrieval algorithm, see also Table 2. Identical instruments are identified by the same colour; \*quantile mapping target over ocean (section 4.2); \*\*quantile mapping target over land.**

Instantaneous precipitation rates from PMW imagers TMI, GMI, and AMSR-E intercalibrated to the SSMI(S) FCDR are also
retrieved with the above-mentioned ANN. As these PMW imagers have a higher resolution compared to SSMI(S), related
footprints of the L1C data are averaged to match SSMI(S) footprints. Additionally, based on Olson et al. (2001), a correction
factor for the level of mixing of convective and stratiform precipitation was developed using polarization differences at the
inter-calibrated 85 GHz (V-H) vs the average 85 GHz Tbs.

### 3.2.2 PNPR-CLIM

The Passive microwave Neural network Precipitation Retrieval for CLIMatological applications (PNPR-CLIM; Bagaglini et
al., 2021) has been developed within the Copernicus Climate Change (C3S) program for deriving instantaneous precipitation
rates from Tbs observed by AMSU-B and MHS cross-track scanning radiometers as provided in the respective FIDUCEO
FCDR v4.1 (Hans et al., 2019) over land and ocean. PNPR-CLIM consists of two ANNs, one for the detection of precipitation
in AMSU-B/MHS Tbs, and the other one for its quantification. Both are trained with high-quality precipitation observations
by the combined GMI and Dual-frequency Precipitation Radar (DPR) instruments onboard the GPM Core Observatory (GPM-
CO) (2B-CMB V06A data; Grecu et al., 2016), collocated with FIDUCEO MHS data. In addition to instantaneous Tbs and

respective differences between selected channels, the inputs to the algorithm are: the scan angle, a global land/sea mask for surface classification, and ERA5 fields (Hersbach et al., 2020) at 0.25° and monthly resolution for the auxiliary input variables,

namely two-metre temperature, freezing level, total precipitable water vapour, snow depth and sea ice concentration. In the presence of deep convection – detected in Tb differences from the 183 GHz sounding channels – the precipitation rates are calibrated to match the distribution of the high-resolution ground-based Multi-Radar-Multi-Sensor System dataset in North America (Zhang et al., 2016), similarly to the quantile mapping applied here (section 4.2). Indicators of the degradation of the quality such as the presence of sea ice or snow (based on ERA5, see also section 4.6) are provided as quality flags. With the

respective FIDUCEO FCDR ending in 2017, PNPR-CLIM was adapted for the NASA PPS L1C MHS observations from 2018 onwards with only negligible discontinuities in spatiotemporally averaged per-satellite precipitation.

**Table 2: Overview over PMW instruments used in GIRAFE v1. Figure 3 illustrates the deployment of these instruments and their usage in GIRAFE v1. Lower maximum latitudes imply orbits further away from sun-synchronicity, hence providing a more diverse diurnal sampling. Co-I = Conical-scanning PMW imager; CT-S = Cross-track scanning PMW sounder; ifo = ice-free ocean only;**
**[†]valid at nadir, distortion at higher scan angles accounted for in the downstream GIRAFE processing; [††]multiples of DPR resolution, no distortion assumed at higher scan angles.**

| Instrument | Type | Retrieval algorithm | Maximum observed latitude | Nominal footprint |
|---|---|---|---|---|
| Special sensor microwave/imager (SSM/I) | Co-I | HOAPS (ifo) | 88° | 28 km x 37 km |
| Special Sensor Microwave Imager / Sounder (SSMIS) | Co-I | HOAPS (ifo) | 89° | 28 km x 45 km |
| Tropical Rainfall Measuring Mission (TRMM) Microwave Imager (TMI) | Co-I | HOAPS (ifo) | 39° | 10 km x 18 km |
| Advanced Microwave Scanning Radiometer - Earth Observing System (AMSR-E) | Co-I | HOAPS (ifo) | 90° | 8 km x 14 km |
| Global Precipitation Measurement (GPM) Microwave Imager (GMI) | Co-I | HOAPS (ifo) | 69° | 8.6 km x 14 km |
| Advanced Microwave Sounding Unit B (AMSU-B) | CT-S | PNPR-CLIM | 90° | 16 km x 16 km [†] |
| Microwave Humidity Sounder (MHS) | CT-S | PNPR-CLIM | 90° | 16 km x 16 km [†] |
| Sondeur Atmosphérique du Profil d'Humidité Intertropicale par Radiometrie (SAPHIR) | CT-S | PRPS | 28° | 10 km x 10 km [††] |
| Advanced Technology Microwave Sounder (ATMS) | CT-S | PRPS | 90° | 16 km x 16 km [††] |

### 3.2.3 PRPS

The Precipitation Retrieval and Profiling Scheme (PRPS) (Kidd et al., 2021) uses L1C observations from PMW sensors
(ATMS and SAPHIR in the case of GIRAFE v1) to retrieve L2 precipitation estimates over both land and ocean. PRPS relies on observational data with minimal, if any, input of ancillary data. PRPS-SAPHIR was developed for operational use alongside

the Goddard Profiling scheme (GPROF; Kummerow et al., 2011) and is obtained from NASA GPM repositories (see section 6). PRPS uses an a priori database which is based upon matched Tb observations of SAPHIR or ATMS and aggregated DPR observations (2B-CMB v06A, see section 3.2.2) with a maximum temporal difference of 5 minutes and geolocation within 2.5

km. For a given SAPHIR or ATMS observation (set of Tbs from different channels), PRPS finds the six closest matched database entries at similar sensor scan positions and averages the respective DPR-based precipitation rates. PRPS-ATMS reveals unrealistically high precipitation rates over ice-covered surfaces, so that respective observations are discarded according to the Operational Sea Surface Temperature and Ice Analysis (OSTIA) at 0.05° resolution (Good et al., 2020) and for static masks over Antarctica, Lake Baikal and Aral Sea. This step is not necessary for PRPS-SAPHIR because of its low-

latitude coverage (Table 2).

## 4 Methods

In this section, the algorithms and their scientific basis for pre-processing and merging of the input data streams is introduced. The GIRAFE algorithm for daily accumulated precipitation (section 4.3) and the respective sampling uncertainty (section 4.4), as well as the upstream infrared quality control (section 4.1) are based on the methods used for the Tropical Amount of

Precipitation with an Estimate of ERrors (TAPEER) algorithm and product (Chambon et al., 2013; Roca et al., 2018), developed for the exploitation of observations by the Megha-Tropiques satellite mission. The pre-processing of PMW-based precipitation rates (section 4.2) is necessary because of the diversity of the PMW database (section 3.2). Sections 4.5 and 4.6 discuss the post-processing steps of aggregating the monthly resolved product and the flag for identifying the quality-reducing presence of surface snow or ice.

**4.1 Infrared quality control**

Except for the MVIRI FIDUCEO FCDR, we use operational geostationary data sets (section 3.1), which require quality control to make them suitable for climate applications. Erroneous scanlines in infrared images are flagged using a radiometric quality control algorithm (Szantai et al 2011, Lorant et al 2017) and are subsequently ignored in the GIRAFE algorithm.

**4.2 Quantile mapping of PMW-based precipitation rate estimates**

The various sources of the PMW-based input stream require a homogenisation procedure. This is particularly important because of the use of a detection threshold (section 4.3), since a mismatch between satellites or retrieval algorithms will have a strong impact on the training of the infrared observations in the final GIRAFE output. Here, quantile mapping is applied, which is a common method for constructing precipitation datasets from a variety of PMW-based input. For example, Huffman et al. (2007), Tan et al. (2021), or Yamamoto and Kubota (2022) use quantile mapping or methods, although the details vary.

We derive the mapping from non-collocated instantaneous (Level-2) precipitation rate estimates and apply it to these same instantaneous observations.

Observations by SSMIS onboard DMSP-F17 (algorithm HOAPS) and MHS onboard METOP-A (algorithm PNPR-CLIM) were selected as targets for the quantile mapping procedure over ocean and land, respectively (Figure 3), because of the long

and stable timeseries from these sensors. Instantaneous precipitation rates $r$ observed by other satellites according to the respective retrieval algorithms are converted to match the distributions of the target platforms:

$$r_T = g^{-1}\big(f(r)\big) \tag{1}$$

Here, $f$ and $g$ are the cumulative probability density functions of the precipitation rates from the mapped and the target satellite, respectively. The mapped precipitation rates, $r_T$, are used in the PMW-based input stream to GIRAFE. As the

distributions representing $f$ and $g$ depend on precipitation regimes and overall climatological situations, they are constructed separately for each month of the year and for each surface type (land and ocean, hence two target satellites) by counting the number of instantaneous observations in respective precipitation rate bins and then accumulating along the precipitation rate dimension. Also, the distributions are collected separately over bands in latitude (8° spacing at the equator to 20° at higher latitudes) and longitude (six bands in total, separating continents). Besides the definition of distributions by month of the year,

no time dependency is implemented, i.e. the distributions are based on all available years from the respective satellite between 2002-2020. This also implies that this method impacts on the homogeneity and stability of the GIRAFE v1 time series only by removing average inter-sensor discontinuities, but not any drifts of single satellites. In the presence of a strong drift, residual discontinuities may occur when the respective satellite joins or leaves the constellation. However, based on the findings in sections 5.3 and 5.4, there is no evidence for such situations when comparing against other (quasi-)global datasets over the

regions specified in our homogeneity and stability analyses.

Months with strong El Nino / Southern Oscillation (ENSO) amplitudes as per the MEI v2 index (Wolter and Timlin, 2011) are discarded during the collection of distributions, due to reportedly high dependency of HOAPS on these conditions (e.g. Masunaga et al., 2019). Strong ENSO events are identified by the MEI v2 index surpassing 1.0 (El Nino) or falling below -

1.25 (La Nina). These thresholds have been defined *ad hoc*.

The mapping as per Eq. (1) is substituted by an identity mapping (i.e., no quantile mapping, effectively) in latitudes above 70°N/S during the hemispheric winter half year, due to few or no (non-zero) observations over snow and ice in either of the three sub-databases (HOAPS, PNPR-CLIM, PRPS, sections 3.2.1-3.2.3), and in latitude/longitude boxes with less than 3% of

the respective surface type (land/ocean), due to few observations possibly leading to spurious distributions. Finally, the mapping is interpolated in narrow transitions between the above-mentioned latitude and longitude bands in order to avoid unphysical discontinuities at these boundaries. The positive homogenising effect of the quantile mapping procedures is illustrated in Appendix A.

Observed Tbs are often affected by geometric distortions, leading among others to a very low spatial resolution, especially in the case of cross-track scanning microwave sounders. Therefore, the five outer scan positions for AMSU-B and MHS are discarded for both the construction of the distributions and the later use in the merging procedures, according to PNPR-CLIM quality flagging. For the other cross-track scanning sensors ATMS and SAPHIR, only the outer two scan positions are discarded because the sensor resolution is prescribed by the PRPS algorithm (Table 2). Microwave imagers are less affected
due to their conical scanning geometry.

Data from single satellites during periods when Tbs and hence precipitation rate estimates are contaminated or show a strong trend not detected by other satellites are discarded for both the quantile mapping and the later use in the merging. These periods are detected in a non-automated fashion by eye in per-satellite anomaly timeseries of monthly means over different regions
(tropics, northern/southern extra-tropics, separately over land and ocean, see Niedorf et al. (2024b)). Such trends occur mostly at the start or towards the end of the lifetime of a satellite. The respective removals lead to the periods detailed in Figure 3, which in these cases deviate from the actual satellite uptimes. Data from a single shorter period of contamination for NOAA-19 in October 2017 as reported by Sanò et al (2021) were also removed. No further quality control was carried out on the PMW-based data. The utilisation of mostly quality-controlled FCDR sources for PMW Tb observations in GIRAFE gives
confidence that contaminations are rare, but their occurrence cannot be entirely excluded.

**4.3 Derivation of daily accumulated precipitation**

The estimation of accumulated precipitation is based on the TAPEER algorithm (Chambon et al., 2013). The merging of PMW and infrared observations inside 55°N/S follows the GOES Precipitation Index technique (Arkin, 1979), more specifically the Universally Adjusted GOES Precipitation Index (Xu et al., 1999), see section 4.3.1. for details. This approach separates the
daily accumulated precipitation ($P_{acc}$) in a 1° x 1° x 1 day (1DD) grid cell into the conditional precipitation rate ($R$, in mm/h) and the precipitation fraction ($F$, unitless) in this cell:

$$P_{acc} = R \cdot F \cdot 24\,\text{h}\,, \tag{2}$$

hence assuming that if it rains in the grid cell, it occurs at a constant average intensity. The derivation of $R$ is based only on the PMW-database:

$$R = \sum_i r_i a_i / \sum_i a_i\,. \tag{3}$$

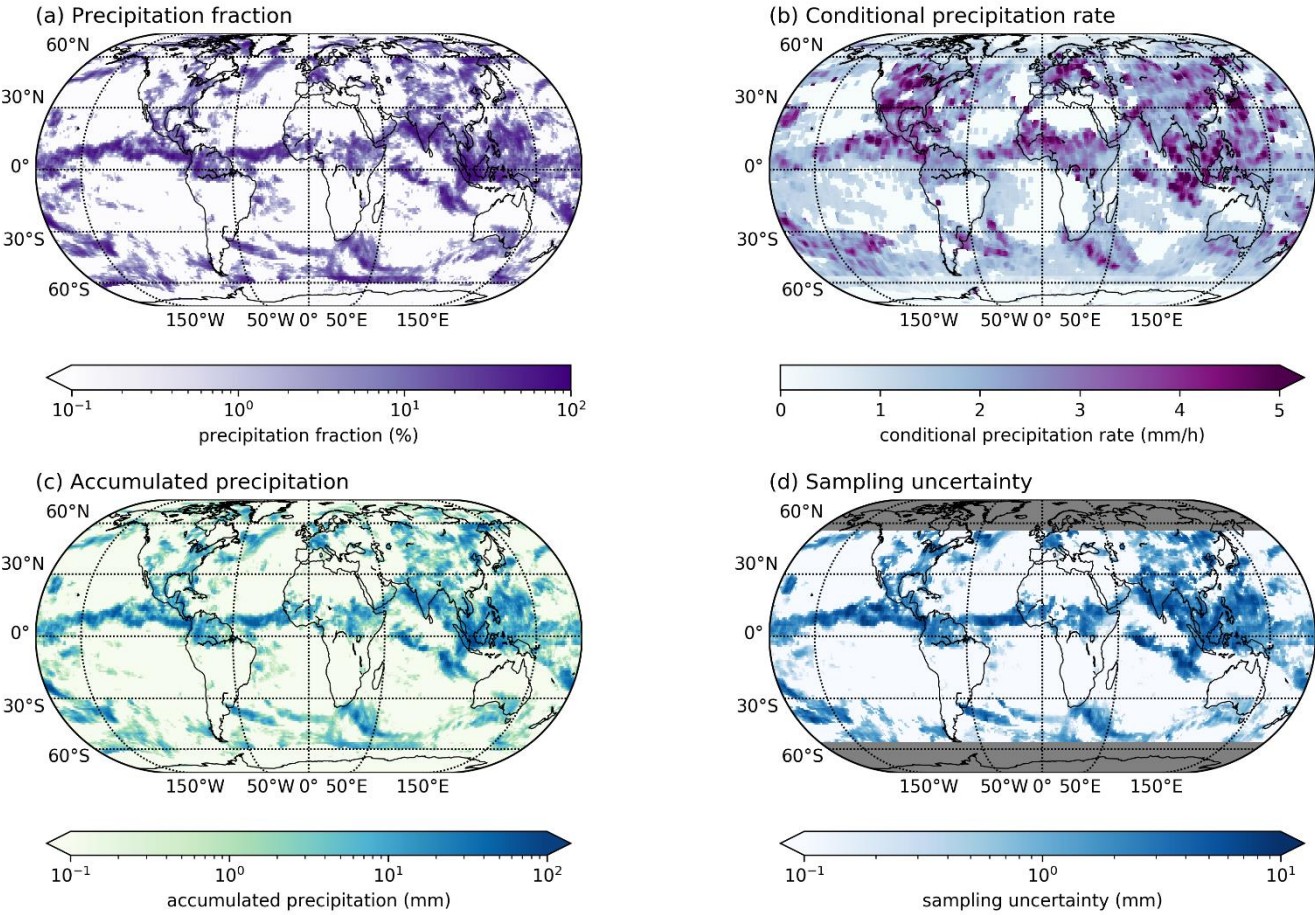

**Figure 4: Example fields for the GIRAFE v1 precipitation fraction $F$ (panel a), the conditional precipitation rate $R$ (panel b), the resulting daily accumulated precipitation $P_{acc}$ (panel c), and the sampling uncertainty $\sigma_S$ (panel d) on July 13th 2021. The minimum values of the fields illustrated in panels a, c, and d using logarithmic color-scales are zero.**


Here, the index $i$ specifies instantaneous PMW observations in the 3° x 3° x 3 days (3DD) environment centered around the 1DD grid cell, regardless of the source satellite/sensor, for which the precipitation rate estimates $r_i$ lie above a pre-defined rate threshold (Table 3). The choice of 3DD for the size of the environment optimizes the frequency distributions of 1DD precipitation rates simultaneously with the respective rate thresholds and detection thresholds (see below) when compared to

established datasets The respective footprint areas $a_i$ (Table 2) are used for linearly weighting the precipitation rate estimates because observations of large areas are expected to be more representative for the entire 3DD environment than those of smaller ones. The derivation of $F$ and the rate threshold for the derivation of $R$ depend on whether the 1DD grid cell is inside (section 4.3.1) or outside of 55°N/S (section 4.3.2).

One day is defined from 00:00:00 UTC to 00:00:00 UTC on the next day in the official product discussed here. However, for users who need to rely on a different local time equivalent, unofficial variants with periods shifted by 6, 12, and 18 hours exist and are available upon request (see section 6). Figure 4 shows the resulting intermediate results for $F$ and $R$ (panels a and b) for one exemplary day, as well as the final daily accumulated precipitation $P_{acc}$ as per Eq. (2) in panel c.

### 4.3.1 Inside 55°N/S (merging)

At latitudes below 55°N/S, where infrared observations from geostationary satellites are utilized, the GIRAFE algorithm derives the precipitation fraction $F$ from the merged infrared and PMW-based data streams following the Universally Adjusted GOES Precipitation Index technique (Xu et al. 1999). For a given 1DD grid cell, the GIRAFE algorithm finds collocated observations in the infrared and PMW-based data streams in a 3° x 3° x 1 day environment centered around the 1DD grid cell. Observations are ingested into the collocation database if i) the infrared observation lies inside the much larger footprint ellipse

of the respective PMW observation (spatial criterion, see Table 2) and if ii) the infrared observation and the PMW observation were recorded within the duration of one geostationary infrared imager scan cycle (temporal criterion, see Table 1). The infrared observations are processed at the sensor-specific resolution and sampling rate; differences between the sensors do not impact the results systematically because of the above-mentioned training that is local in space and time.

Based on this collocation database, a threshold for the detection of precipitation in the infrared Tbs is derived ("infrared threshold", Table 3) for the 1DD grid cell. For this, the fraction of PMW-based precipitation rate estimates exceeding the detection threshold of 0.5 mm/h (Table 3) in the collocation database is determined. The infrared threshold is then derived as the quantile in the infrared Tb distribution in the collocation database, which corresponds to this PMW-based precipitation fraction, cf. Figure 5. The precipitation fraction, $F$, is computed as the fraction of precipitation events (Tb below the derived

infrared threshold) in all infrared observations falling into the 1DD grid cell.

Inside 55°N/S, the conditional precipitation rate for a 1DD cell is computed using a rate threshold of 1.0 mm/h, which was found to bring the overall frequency distributions of daily accumulated precipitation in GIRAFE v1 closest to established datasets in early results – in tandem with the 0.5 mm/h detection threshold used for the training of the infrared threshold (see

above) and the 3DD environment for the averaging of the conditional precipitation rate.

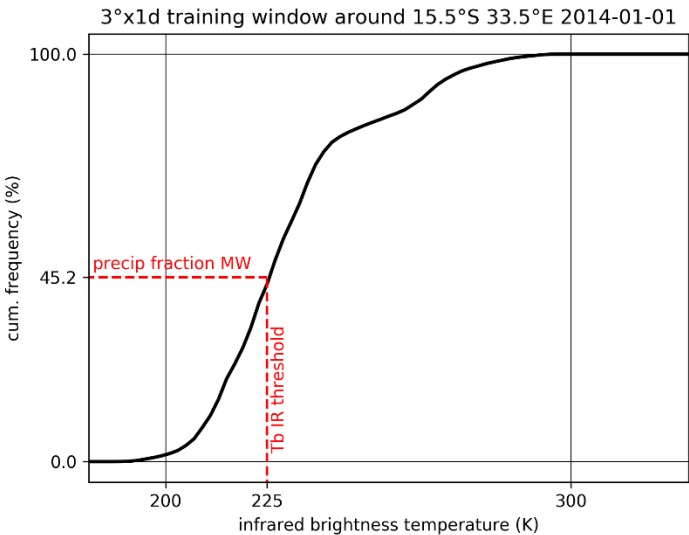

**Figure 5: Example derivation of the infrared threshold (vertical red dashed line) based on the PMW-based precipitation fraction (horizontal red dashed line) and the distribution of infrared (IR) Tbs (thick black line), both in the collocation database of the respective 1DD grid cell.**

**Table 3: Overview over the thresholds used for the computation of the conditional precipitation rate *R* and the precipitation fraction *F* during the merging in the GIRAFE algorithm.**

| Name | Application | Value inside 55°N/S | Value outside 55°N |
|---|---|---|---|
| Rate threshold | Filter PMW-based data stream for observations contributing to the conditional precipitation rate derivation | 1 mm/h | 0.5 mm/h |
| Detection threshold | Find precipitation events in the PMW-based data stream | 0.5 mm/h | 0.3 mm/h |
| Infrared threshold | Find precipitation events in the infrared data stream | Trained locally by PMW-based observations using the collocation database | Not applicable |

## 4.3.2 Outside 55°N/S

Where no infrared data are used, the GIRAFE algorithm relies only on the PMW-based data stream as follows: The precipitation fraction *F* is the fraction of PMW-derived precipitation rates exceeding 0.3 mm/h in the 1DD grid cell (Table 3). This threshold for detection of precipitation in the PMW data stream is lower than inside the 55°N/S region (0.5 mm/h, see section 4.3.1) due to precipitation occurring at lower rates in higher latitudes in general. In this respect, GIRAFE deviates from

the TAPEER methods established by Chambon et al. (2013) who did not consider regions of the planet where infrared observations from geostationary satellites cannot be used.


The 1DD conditional precipitation rate is computed in the same way as inside 55°N/S, but using a rate threshold of 0.5 mm/h (Table 3), which proved optimal in reproducing frequency distributions of established datasets in tandem with the 0.3 mm/h detection threshold and the 3DD environment for the averaging of the conditional precipitation rate (see above) in an early version.

**4.4 Daily sampling uncertainty**

The GIRAFE sampling uncertainty is based on instantaneous precipitation fields which are obtained from the infrared Tb observations by applying the 1DD infrared threshold as explained in section 4.3.1. It is therefore only available inside the 55°N/S latitude band covered by geostationary satellites. These instantaneous precipitation fields can take two different values: 0 for infrared Tbs exceeding the infrared threshold (no precipitation) or the conditional precipitation rate $R$ otherwise.

Following Roca et al. (2010) and Chambon et al. (2013), the sampling uncertainty, $\sigma_S$, is computed as

$$\sigma_S = \frac{\sigma}{\sqrt{N_{ind}}}, \tag{4}$$

with $\sigma = R \cdot \sqrt{F \cdot (1 - F)}$ being the standard deviation of the binary infrared-based precipitation field in the 1DD cell. With strong correlations present between neighbouring infrared pixels, the number of independent observations $N_{ind}$ in Eq. (4) is well below the number of available infrared observations. It is estimated as

$$N_{ind} = \frac{A \cdot T}{d^2 \cdot \tau}, \tag{5}$$

where $A$ is the area of the 1° x 1° grid cell, $T$=24 h is the temporal extent of the grid cell, and $d$ and $\tau$ are the decorrelation scales of the precipitation signal in space and time, respectively. These decorrelation scales are estimated separately from exponential fits to variograms of the binary infrared-based precipitation fields in the respective dimensions.

This uncertainty measure is useful when satellite and ground-based estimates are compared as detailed for instance in Roca et 325 al. (2010) and Gosset et al. (2018). It has also been successfully used to assess the sensitivity of the satellite products to the configuration of the microwave constellation (Roca et al. 2018; Oliveira and Roca 2022). Dataset-specific uncertainties could also be very useful in the context of hydrological modelling applications, as an added value for exploring different scenarios of river discharge simulations through ensembles constructed by otherwise offline estimates of precipitation uncertainty (e.g., Paiva et al., 2013, Wongchuig et al., 2024), particularly in intertropical basins, which are subject to strong spatiotemporal 330 variability of precipitation.

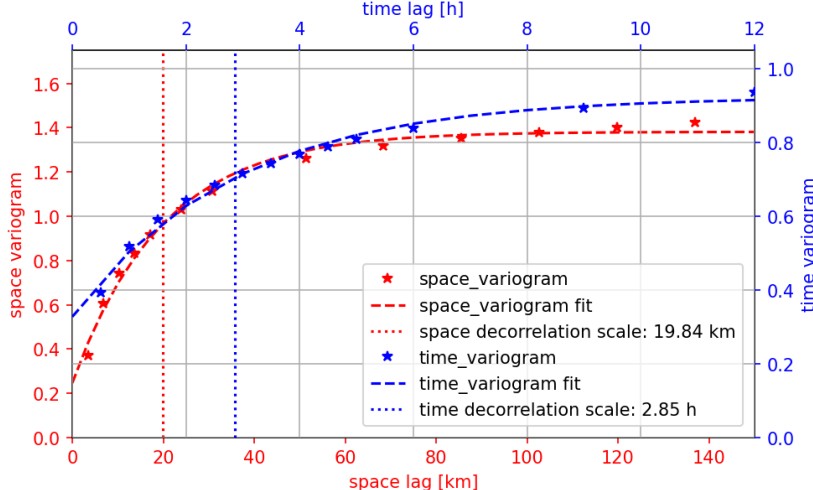

**Figure 6: Example space variogram (red) and time variogram (blue) of a 5° x 5° x 10 days environment. The 5° x 5° box is centred at 12.5°N / 27.5°W (tropical East Atlantic Ocean), the time period covers July 11th to July 20th 2021. Red axes (left and bottom) belong to the space variogram, blue axes (right and top) are assigned to the time variogram. Stars mark the computed experimental variograms, dashed lines show the respective fit. The dotted vertical lines illustrate the fitted decorrelation scales in space (blue, 19.84 km) and time (red, 2.85 hours).**

The variograms are collected over larger fixed (i.e., non-moving) 5° x 5° x 10 days environments, thereby increasing the underlying databases to form a larger statistical ensemble rather than covering only few events. The variograms in spatial dimensions are first computed per scan in 5° x 5° cells based on spatial increments in scan line and scan position directions only, and then the per-scan variograms over the 10-day period are averaged. Conversely, the temporal variograms are first computed per geostationary pixel over the 10-day period based on differences in scan times, and then the per-pixel variograms are averaged over the 5° x 5° boxes. For both types of the variograms, the binary infrared-based precipitation fields are sampled to coarser resolutions, depending on sensor specifications (scan frequency and spatial resolution at nadir, see Table 1), roughly matching a temporal resolution of 1 h and a spatial resolution of 5 km (nadir) for the spatial variograms and 30 minutes and 15 km for the temporal variograms, respectively. In general, this resampling harmonizes the available resolutions of different sensors as far as possible. However, the lowest resolution is not always taken as target to not reduce the quality of the more advanced instruments. Therefore, outliers such as the VISSR sampling frequency of 60 minutes cannot be overcome and the resulting larger minimum lag between two observations for e.g. VISSR may have a systematic effect on the resulting decorrelation scales. Figure 6 shows example space variogram and time variogram of a 5° x 5° x 10 days environment, its respective fits and decorrelation scales.

The fit procedure works well for truly exponentially shaped variograms, such as the ones shown in Figure 6. There are situations in which the variograms cannot be calculated or have irregular shapes, for example falling below the plateau at higher lags or an alternation of high and low values at low lags instead of a monotonous increase, possibly due to many missing infrared scenes or in arid settings. This typically leads to implausibly low or high or entirely missing decorrelation scales, due

to an numerical instabilities in the fitting routine in the latter case. These situations are filtered by identifying cases for which the decorrelation scale is below the minimum lag, above the maximum lag, or not retrieved. In order to avoid gaps, the respective values are replaced by climatological values (20 km for spatial and 1.5 h for temporal decorrelation). These values are relatively well aligned with the most frequent values of the variogram fits in both GIRAFE v1 and TAPEER v1.5 (Figure 5-13 in Konrad et al. (2024)). A flag is provided which identifies these situations and allows the rejection of respective derived uncertainty estimates.

Figure 4 shows the resulting sampling uncertainty as per Eq.s (4) and (5) for one exemplary day in panel d.

## 4.5 Monthly aggregation of daily accumulated precipitation

For the convenience of users who are interested in an average monthly precipitation, the global 1DD accumulated precipitation fields (section 4.3) are averaged along the temporal dimension over each month, yielding 1° x 1° x 1 month (1DM) gridded global fields. The number of days available for averaging is provided for each 1DM grid cell, as well as a flag which informs users whether there are more than ten days or more than four consecutive days missing in the 1DM grid cell, in which case the 1DM value is non-compliant with guidelines by the World Meteorological Organization (WMO, 2017). The daily sampling uncertainty is not propagated to the 1DM resolution, due to the unknown day-to-day uncertainty correlation.

## 4.6 Snow/ice flag

With HOAPS not providing PMW-derived precipitation rates over ice- or snow-covered areas, PRPS output being specifically filtered for these situations (section 3.2.3), and PNPR-CLIM being reportedly weak in these situations (Bagaglini et al., 2021), a snow/ice flag (SIF) is introduced following the PNPR-CLIM quality flag setting for snow and ice cover: ERA5 snow depth and sea-ice concentration (Hersbach et al., 2020) above zero in a 3DD environment (matching the maximum extent of the GIRAFE environments) around a 1DD grid cell indicate a probable degradation of the data quality in GIRAFE v1, hence SIF=1, otherwise SIF = 0 for the 1DD grid cell. For the monthly GIRAFE v1 data, the number of days with SIF = 1 are counted for each 1DM grid cell.

## 5 Validation, intercomparison, and verification

GIRAFE v1 comes with an extensive validation report (Konrad et al., 2024), featuring comparisons against reference datasets and similar quasi-global datasets, a dedicated stability and homogeneity analysis, the verification of the plausibility of the uncertainties and the underlying decorrelation scales, and an analysis of missing values in the 1DD and 1DM datasets. In this section, we condense the information of this validation report and refer the interested reader to the validation report for the extended analysis. Particular emphasis is put on the homogeneity analysis (section 5.3), the stability analysis (section 5.4) and the analysis of the scaling of daily extreme precipitation with sea surface temperature (SST) which has not featured in the

validation report (section 5.5). Finally, section 5.6 illustrates how the GIRAFE v1 sampling uncertainty resembles the validated TAPEER v1.5 uncertainties.

We focus on the assessment of the 1DD dataset of the GIRAFE v1 CDR. In order to put the GIRAFE v1 CDR results in
perspective, the validation activities presented in section 5.1 are carried out also for the established (quasi-)global satellite-based datasets: Global Precipitation Climatology Project (GPCP) v3.2 (Huffman et al., 2023a), Climate Prediction Center Morphing Technique (CMORPH) v1 (Xie et al., 2017), Integrated Multi-satellitE Retrievals for GPM (IMERG) v6 (Huffman et al., 2020) and v7 (Huffman et al., 2023b, not present in the previously mentioned validation report by Konrad et al., (2024)), Global Satellite Mapping of Precipitation (GSMaP) NRT v8 (Kubota et al., 2007; not present in the previously mentioned
validation report by Konrad et al. (2024)). The reanalysis dataset ERA5 (Hersbach et al., 2020) is also included in these comparisons. GPCP v3.2 is corrected towards rain gauge observations over land, which is not the case for GIRAFE v1. There are both corrected and uncorrected versions for CMORPH (CRT and RAW), IMERG (FC and FU), and GSMaP (with and without "gauge" in the labels below). Where feasible, we also include TAPEER v1.5 (Roca et al., 2018). For the homogeneity and stability analyses (sections 5.3 and 5.4), we also use the TRMM Multi-satellite Precipitation Analysis (TMPA) 3B42 v7
(Huffman et al., 2016) and the Global Precipitation Climatology Centre (GPCC) v2022 (Ziese et al., 2022) datasets. All datasets are retrieved at 1DD resolution from the Frequent Rainfall Observations on GridS (FROGS) archive (Roca et al., 2019).

Following for example Gosset et al. (2018), we rely on established indicators of quality for assessing the various datasets, namely bias (mean difference), bias-corrected root-mean-square difference (bc-RMSD), correlation coefficient (CC), detection
statistics (hit rate, HR; probability of detection, POD; false alarm rate, FAR; and Heidke Skill Score, HSS), and the frequency of error bar overlap (FEBO). Apart from the detection statistics, no distinction is made between non-zero and no precipitation during the computation of the indicators. Where spatial dimensions are collapsed during the computation of the bias and bc-RMSD, each 1DD grid cell is weighted according to the area, i.e., proportional to sin(latitude). For the detection scores, the occurrence of precipitation in a 1DD grid cell is determined at 1 mm/d which is a common threshold for distinguishing dry
and rainy days (e.g. Gosset et al., 2018).

## 5.1 Validation

The analyses in sections 5.1.1 and 5.1.2 are considered validations in the sense that the validating local dataset AMMA-CATCH and regional dataset EURADCLIM are high-resolution and high-accuracy. These datasets are limited both in time and more so in space. Already between these two regions, the advantages and disadvantages of the validated datasets vary (see
below). It can be expected that regions with other surface settings and climatological conditions will also lead to different results. Hence, the validation here can only be considered a sample rather than complete.

### 5.1.1 AMMA-CATCH

We use the data from a high-resolution, dense rain gauge network near Niamey, Niger, in the "African Monsoon Multidisciplinary Analysis – Couplage de l'Atmosphère Tropicale et Cycle Hydrologique" (AMMA-CATCH) dataset (Lebel et al., 2009) to validate GIRAFE v1 and the other above-mentioned (quasi-)global datasets. AMMA-CATCH has been used previously for the validation of satellite-based precipitation estimates (Gosset et al., 2018 and references therein). The Niamey gauge network covers the 1° x 1° area around 13.5°N, 2.5°E and consists of 40-50 stations. Accumulated precipitation from these stations is extended to the entire 1° x 1° area by kriging before aggregation over spatial dimensions. The respective kriging uncertainty is available. We use 24-hour accumulations from 2002-2019. Precipitation in Niamey is governed by the West African Monsoon, so the comparison is carried out for observations between June and September. It is noted that according to the above specification of the Niamey network, the validation extends to only a single grid cell in spatial dimensions for GIRAFE v1.

**Table 4: Statistics from the 1DD comparison of the evaluated datasets against the AMMA-CATCH Niamey data. The comparison is carried out for June-September in 2002-2019 (exceptions: [†]2002-07/2017; [‡]2012-2019).**

| Dataset | Bias (mm/d) | bc-RMSD (mm/d) | CC (%) | HR (%) | POD (%) | FAR (%) | HSS (%) | FEBO (%) |
|---|---|---|---|---|---|---|---|---|
| Datasets without adjustment towards rain gauge data | | | | | | | | |
| GIRAFE v1 | 1.10 | 5.67 | 75 | 83 | 92 | 24 | 66 | 60 |
| CMORPH v1 RAW[†] | 2.24 | 7.12 | 85 | 89 | 91 | 13 | 78 | 43 |
| TAPEER v1.5[‡] | -0.16 | 3.80 | 88 | 90 | 87 | 8 | 79 | 73 |
| IMERG v6 FU | 1.76 | 6.00 | 89 | 91 | 94 | 12 | 81 | 47 |
| IMERG v7 FU | 0.95 | 5.31 | 87 | 90 | 91 | 11 | 80 | 46 |
| GSMaP NRT v8 | -0.37 | 6.32 | 72 | 86 | 78 | 9 | 70 | 46 |
| Datasets with adjustment towards rain gauge data | | | | | | | | |
| GPCP v3.2 | -0.17 | 3.60 | 90 | 92 | 91 | 8 | 83 | 53 |
| CMORPH v1 CRT | 0.31 | 4.75 | 85 | 89 | 87 | 9 | 78 | 52 |
| IMERG v6 FC | 0.05 | 3.54 | 91 | 91 | 92 | 9 | 83 | 51 |
| IMERG v7 FC | 0.01 | 3.76 | 90 | 91 | 91 | 9 | 81 | 48 |
| GSMaP Gauge NRT v8 | -0.05 | 5.99 | 78 | 87 | 82 | 8 | 74 | 47 |
| Reanalysis | | | | | | | | |
| ERA5 | -1.78 | 8.30 | 35 | 67 | 52 | 22 | 30 | 21 |

GIRAFE v1 has a bias and bc-RMSD against AMMA-CATCH which is in the range of other datasets which have not been adjusted to rain-gauge datasets (Table 4). While the POD of GIRAFE v1 is as high as in most other datasets, the GIRAFE v1

FAR is notably high, leading to lower scores also in terms of CC, HR, and HSS. The overestimation of precipitation in GIRAFE v1 is likely a feature of the PNPR-CLIM Level-2 data which also produces increased precipitation over Africa (Bagaglini et al., 2021). The datasets which are adjusted to rain-gauge datasets usually perform better, indicating that future developments in GIRAFE context involving such an adjustment may also improve these scores. ERA5, except for the FAR statistic, performs weakest against the AMMA-CATCH dataset, possibly because no information relevant to precipitation is assimilated in the vicinity of the Niamey 1DD grid cell. In the absence of 1DD uncertainty information in the other datasets, except TAPEER v1.5, the FEBO of GIRAFE v1 is exceptionally high (60%). The better performance of TAPEER v1.5 in this metric is likely a result of this dataset being designed for tropical conditions, of the generally lower bias and of the better detection statistics. A more detailed discussion of the sampling uncertainty is given in section 5.6.

### 5.1.2 EURADCLIM

We compare GIRAFE v1 and the above mentioned (quasi-)global datasets to the 2013-2020 ground-based precipitation radar dataset EURADCLIM (Overeem et al., 2023). The 24 h accumulations are averaged from their native 2 km x 2 km resolution to 1DD. An *ad hoc* sampling uncertainty is computed according to Eq.s (4) and (5), excluding the temporal dimension and with the default GIRAFE spatial decorrelation scale of 20 km. With the temporal sampling not being accounted for, the resulting uncertainty is likely to be underestimating the true sampling uncertainty in EURADCLIM. The comparison is restricted to 1DD grid cells south of 60°N for a fair comparison with CMORPH v1 and IMERG which are (mostly) not available north of this latitude. The 528 remaining 1DD grid cells cover West and Central Europe, southern Scandinavia and the southern Baltic Sea, and parts of East Europe, see Appendix B for an illustration of the availability of mutual grid cells.

GIRAFE v1 tends to underestimate the amount of precipitation in EURADCLIM (low bias shown in Table 5), but remains largely in the range of the other unadjusted datasets. The overall bc-RMSD given in Table 5 is very close to the optimal values found in all datasets. This finding is confirmed in the comparison of spatial distributions of the bc-RMSD (Figure 7). In general, GIRAFE v1, like the other datasets, sees an increased deviation from EURADCLIM over the Alps, the Pyrenees, the Balkans, the northern parts of Ireland and Great Britain, and the southern Norwegian coast. Most of these areas are mountainous, which makes the inference of precipitation from both satellites and ground-based radars more difficult.

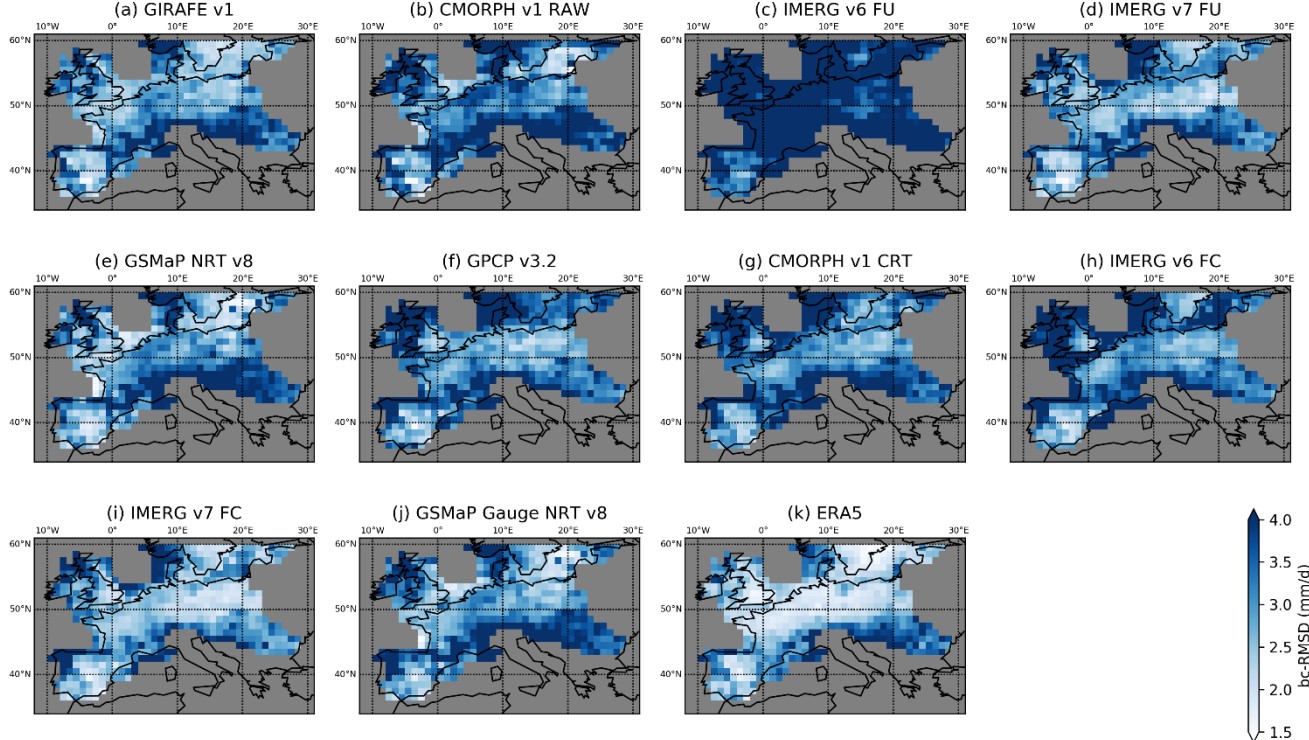

**Figure 7 Maps of the bc-RMSD in the 1DD precipitation for the validated datasets against EURADCLIM. The datasets appear in the same order as in Table 5. Panels (a)-(e) represent the unadjusted datasets and, (f)-(j) represent the adjusted datasets, and panel (k) represents the ERA5 reanalysis. GIRAFE v1 has been analysed without the application of the snow flag. The analysis of CMORPH v1 RAW (panel (b)) ends in July 2017; all other comparisons extend over mutually available grid cells in the 2013-2020 period.**

The detection statistics of GIRAFE v1 (and hence also CC) suffer from a relatively low POD (Table 5). Other unadjusted datasets have similarly low or lower POD, while the adjustment towards rain gauges for most datasets alleviates this effect. The evaluation of GIRAFE v1 over summer months only (row 2 in Table 5) yields a significant improvement in all measures towards or above the other datasets all-year performance, while the evaluation of GIRAFE v1 over the entire year, but with

470 the SIF (section 4.6) applied ranges in-between, pointing to the low POD being related to detecting snowfall or liquid/mixed precipitation over snow- and ice-covered regions. It should be noted that the application of the SIF removes ~45% of the 1DD grid cells in the EURADCLIM-GIRAFE v1 comparison, due to the conservative extension of the ERA5 information on snow and sea ice cover to a 3DD environment (section 4.6). Again, the adjusted datasets score better in principle than their unadjusted counterparts, indicating the potential of a future adjustment in GIRAFE. Finally, as in section 5.1.1, in the absence of 1DD

uncertainty estimates in the other datasets, GIRAFE v1 outperforms these in terms of FEBO; a more detailed discussion of the sampling uncertainty can be found in section 5.6.

**Table 5: Statistics from the 1DD comparison of the evaluated datasets against the EURADCLIM dataset. Only latitudes below 60°N are considered. The comparison is carried out for 2013-2020 (exception: [†]2013-07/2017).**

| Dataset | Bias (mm/d) | bc-RMSD (mm/d) | CC (%) | HR (%) | POD (%) | FAR (%) | HSS (%) | FEBO (%) |
|---|---|---|---|---|---|---|---|---|
| Datasets without adjustment towards rain gauge data | | | | | | | | |
| GIRAFE v1 | -0.86 | 3.17 | 74 | 82 | 62 | 5 | 60 | 52 |
| GIRAFE v1 (June-August) | -0.31 | 3.05 | 78 | 86 | 74 | 6 | 70 | 60 |
| GIRAFE v1 (SIF applied) | -0.55 | 2.89 | 77 | 85 | 66 | 5 | 66 | 57 |
| CMORPH v1 RAW[†] | -1.36 | 3.62 | 64 | 74 | 41 | 3 | 42 | 39 |
| IMERG v6 FU | 0.45 | 5.12 | 73 | 85 | 74 | 8 | 68 | 40 |
| IMERG v7 FU | -0.48 | 3.19 | 77 | 83 | 68 | 6 | 64 | 40 |
| GSMaP NRT v8 | -0.72 | 3.36 | 73 | 82 | 63 | 6 | 60 | 39 |
| Datasets with adjustment towards rain gauge data | | | | | | | | |
| GPCP v3.2 | 0.06 | 3.68 | 76 | 84 | 72 | 8 | 65 | 42 |
| CMORPH v1 CRT | -0.62 | 3.71 | 69 | 79 | 54 | 6 | 51 | 39 |
| IMERG v6 FC | 0.25 | 4.00 | 78 | 85 | 75 | 8 | 69 | 41 |
| IMERG v7 FC | -0.03 | 3.01 | 83 | 85 | 75 | 8 | 68 | 49 |
| GSMaP Gauge NRT v8 | -0.55 | 3.37 | 74 | 82 | 65 | 7 | 61 | 38 |
| Reanalysis | | | | | | | | |
| ERA5 | -0.04 | 2.83 | 82 | 87 | 82 | 10 | 72 | 40 |

## 5.2 Intercomparison with (quasi-)global datasets

GIRAFE v1 features the expected climatological patterns over the globe (Figure 8, panel a vs. b-g). It tends to overestimate precipitation with respect to the other datasets in the tropical rain belt and in the western Pacific Ocean (Figure 8h), except in South America. At higher latitudes, GIRAFE v1 generally tends to underestimate the other datasets, with a few exceptions such as western North America or the North American Atlantic coast. This general underestimation is at least partially related to the inadequate detection of precipitation over snow and ice surfaces discussed in section 5.1.2 and shows up strongest in the sub-polar Antarctic where GIRAFE v1 misses most precipitation over sea ice. The different treatment of 1DD grid cells inside and outside of 55°N/S in GIRAFE v1 (section 4.3) manifests in latitudinal discontinuities at the boundaries (Figure 8a and g, also in Figure 11).

As an example of GIRAFE v1's temporal evolution compared to these other datasets, we focus on low latitudes. Again, the validation report (Konrad et al., 2024) contains a more detailed picture for other regions, too. Over land, on average, GIRAFE

v1 resembles the other datasets closely (Figure 9a), except CMORPH v1 and GSMaP v8, which are biased low compared to the other datasets. Over low-latitude ocean (Figure 9b), all datasets show a relation to ENSO, with precipitation being highest during El Nino events. GIRAFE v1 sees the highest precipitation in this region (see also the discussion of Figure 8h above), and also the strongest link to ENSO. This is most likely inherited from the HOAPS PMW dataset, which has been shown to be more sensitive than others (Masunaga et al., 2019).

Without a clear reference, the most obvious deviations of GIRAFE v1 from the other datasets over tropical oceans in terms of more precipitation and a stronger correlation with ENSO is currently classified as a feature rather than a deficiency.

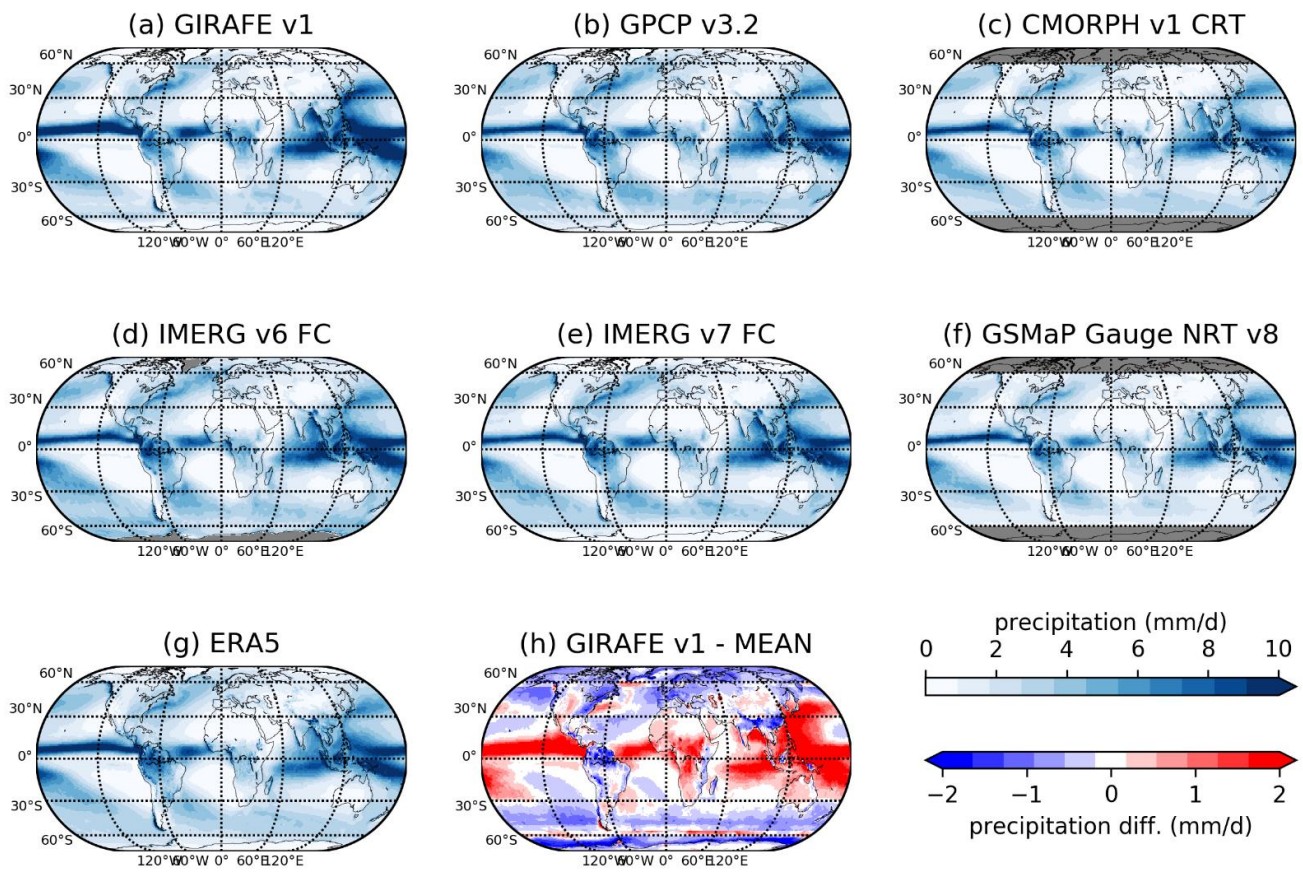

**Figure 8: Spatial distributions of climatological mean precipitation (2002-2020) for various datasets (a-g); panel h: difference between GIRAFE v1 climatological mean (panel a) and the ensemble mean of the other datasets (panels b-g).**

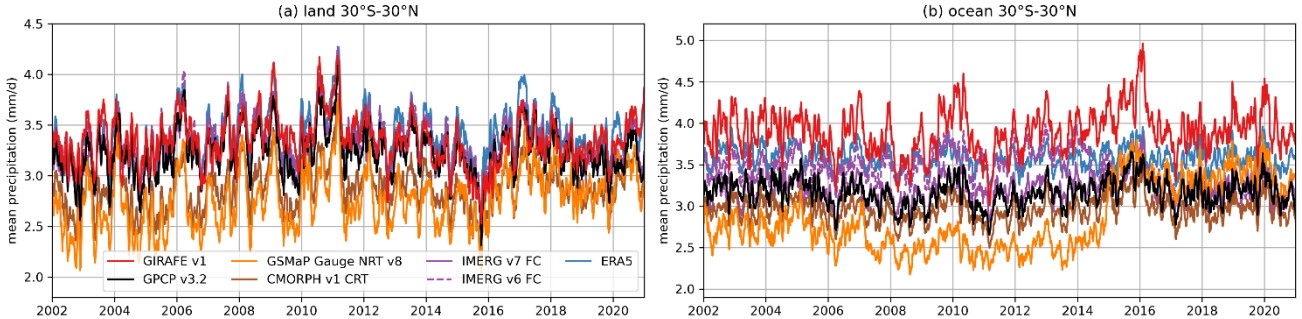

**Figure 9: Timeseries of mean precipitation inside 30°N/S separately for land (a) and ocean (b) for various datasets. The original daily values are all illustrated as thin grey lines; the thick colored lines are obtained via application of a 30-day running average.**

### 5.3 Homogeneity analysis

Homogeneity is an important aspect in a dataset, that is designed for climate applications, as inhomogeneities, or breakpoints, stemming from changes in the observing system can be misinterpreted as a climate signal. In this section, we assess the degree of homogeneity of GIRAFE v1 and other datasets in terms of the occurrence of breakpoints, following methods by Weatherhead et al. (1998) and Mieruch et al. (2014), also utilized in GEWEX water vapour assessment (Schröder et al., 2016, 2019 and Trent et al., 2024). Various homogeneity tools exist (see Venema et al., 2012 for an overview). We apply two tests: the Penalised Maximal F (PMF) test (Wang, 2008a, b) and a variant of the standard normal homogeneity (SNH) test (Hawkins, 1977; Alexandersson, 1986), as proposed in Reeves et al. (2007). The SNH test is carried out only on breakpoints previously identified by the PMF test. The detection of a breakpoint confirmed by the SNH test can then be considered to be of increased confidence because of the two-fold evaluation. These situations are referred to as "confirmed breakpoints" in the discussion below. The resulting breakpoint analysis detects abrupt changes in a time series of precipitation at a 0.05 significance level, in terms of the timing and the strength of the breakpoint.

Input to the PMF and SNH tests are timeseries of anomaly differences, i.e., the difference between the anomalies from a dataset and a reference dataset after removal of the mean annual cycles. Using a record as reference does not constitute a statement on superior quality. Instead, references are chosen because they are widely used, have global (land) coverage and are observation-based (GPCP v3.2, GPCC v2022) or are independent from the former datasets (3B42 v7 after 2000, CMORPH V1 CRT which uses the rain gauge-based dataset CPC (Chen et al., 2008)). For each region and analysed parameter (rows in Figure 10), two reference datasets are chosen to further increase confidence in breakpoints (columns in Figure 10): The occurrence of simultaneous breaks against both reference datasets points to a homogeneity issue in the dataset under consideration. Conversely, mutual breakpoints in several datasets against the same reference dataset may indicate an inhomogeneity in the respective reference dataset.

The anomaly differences analysed in these tests are based either on monthly totals (Figure 10, a-d) or – representing extreme
precipitation (see also section 5.5) – on the monthly 99.9 percentile (Figure 10e, f), each transferred to log-scale, and each
based on the 1DD GIRAFE v1 product and FROGS variants of the other datasets. If a grid-based time series of daily
accumulated precipitation does not cover all days, the monthly total based on the available daily values is scaled accordingly.
More sophisticated approaches use for example reference climatologies (Wang et al., 2023). Finally, it is noted that the
breakpoint analysis is affected by uncertainties; in particular, actually existing breakpoints might go undetected.


Figure 10 shows the anomaly times series and associated breakpoints in the various settings for GIRAFE v1 and several of the
previously introduced datasets. For a detailed overview, Appendix C provides a list of all breakpoints shown in the various
panels. Here, we focus on the discussion of the homogeneity of GIRAFE v1; however, some implications of breakpoints in
the other datasets are discussed in section 5.4. Over global ocean surfaces inside 50°N/S, GIRAFE v1 does not exhibit
confirmed breakpoints (no red star-shaped markers in panels a and b of Figure 10) while over land inside 50°N/S a single,
confirmed breakpoint is detected in December 2008. This breakpoint is only present relative to GPCC v2022 but not relative
to CMORPH v1 CRT (red star-shaped marker in panel c but not in panel d). This reduces confidence in the presence of a
breakpoint in GIRAFE v1 and points to a potential issue in GPCC v2022, which is generated from various input streams of
rain gauge data and intense and – to a significant extent – manual quality control. The combination of this leads to a delay in
the ingestion of data and causes a strong, abrupt decrease in the number of input data in approximately 2009 (see Figure 1 in
Schneider et al., 2022), coinciding with the observed break in GIRAFE v1 and those of other datasets against GPCC v2022 in
October 2008 (GPCP v3.2 and ERA5, see also the full list of breakpoints in Appendix C). Note that consistent with this, GPCC
v2022 together with 3B42 v7 exhibits a break point relative to CMOPRH v1 in October 2008 (again, see also Appendix C).
Further analysis is needed to better understand this feature. GIRAFE v1 does not exhibit confirmed breakpoints relative to
GPCP v3.2 and 3B42 v7 in terms of the monthly 99.9 percentile inside 30°N/S (no red star-shaped marker in panels e and f)
and it is therefore concluded that GIRAFE v1 proves stable in terms of extremes.

## 5.4 Stability analysis

Breakpoints as discussed in section 5.3 or artificial trends caused by drifts or ageing of the observing system itself rather than
the Earth system affect the stability of a CDR. Again following Weatherhead et al. (1998), Mieruch et al. (2014), Schröder et
al. (2016, 2019), and Trent et al. (2024), we define stability as the trend in a timeseries of the bias of one dataset relative to
another one over a given spatial area and assess the stability of GIRAFE v1 in this sense. Trends in the single datasets were
computed as well, utilising the same methods, but applied to absolute values rather than differences (bias). A linear trend
model, the amplitudes and frequencies of four modes, and the strength of ENSO were simultaneously fitted to the timeseries
analysed in section 5.3. The uncertainty of the linear trend estimates was corrected for autocorrelation of autoregressive data
of order 1 (Schröder et al., 2019). The significance of the stability being different from 0 mm/d/decade at a level of 0.05 is
assessed as well. The trends and the stability are estimated for the same cases as in the homogeneity analysis (section 5.3;

monthly totals inside 50°N/S separately for ocean and land and the 99.9 percentile inside 30°N/S). Results are shown in Table 6.

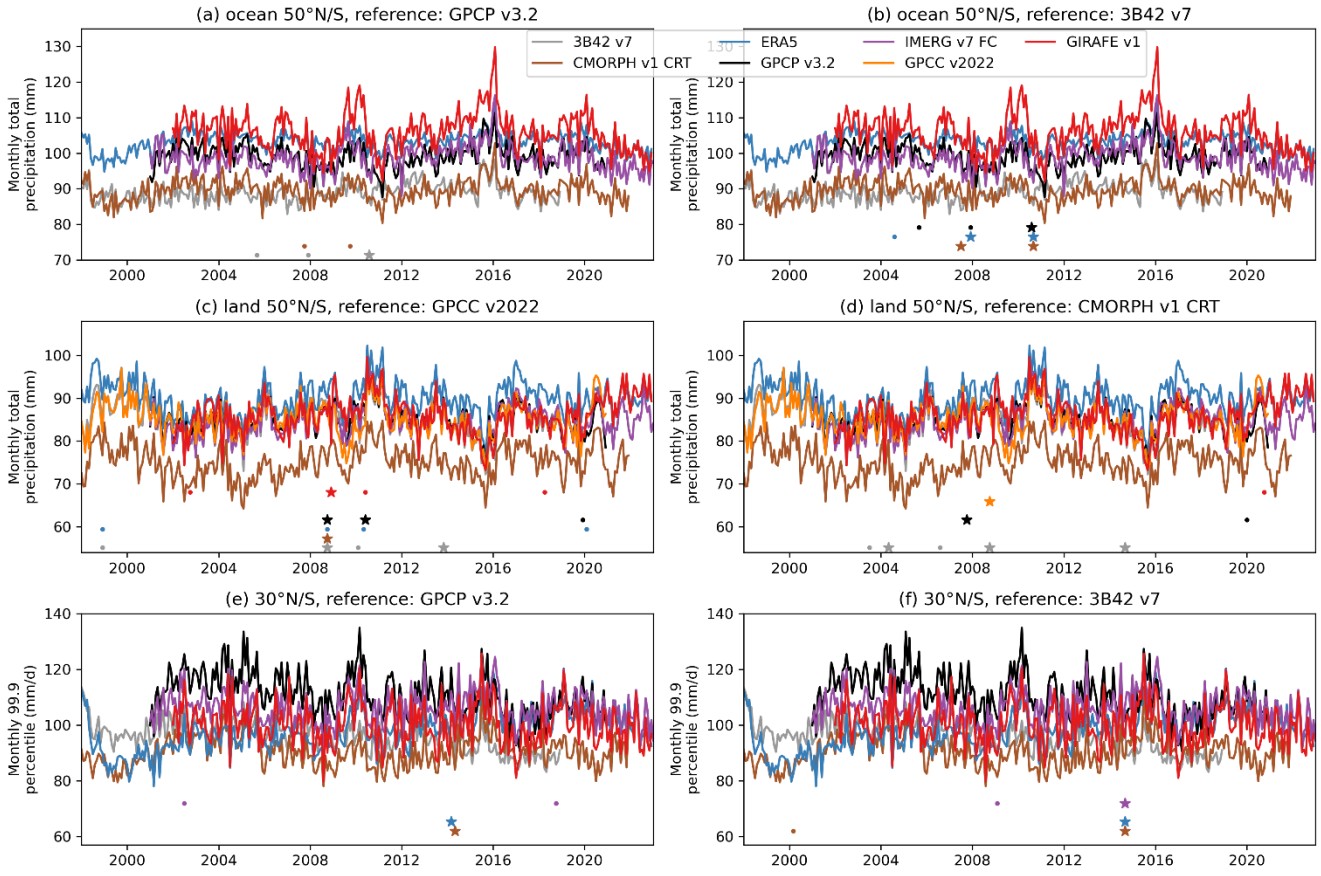


**Figure 10: Homogeneity analysis. (a) Monthly total precipitation over ocean in 50°N/S with GPCP v3.2 as reference dataset for the homogeneity tests. (b) Same with 3B42 v7 as reference dataset. (c) Monthly total precipitation over land in 50°N/S with GPCC v2022 as reference dataset for the homogeneity test. (d) Same with CMORPH v1 CRT as reference dataset. (e) 99.9 percentile of precipitation over land and ocean in 30°N/S with GPCP v3.2 as reference dataset for the homogeneity test. (f) Same with 3B42 v7 as**
**reference dataset. The timeseries (identical in left and right column per row) illustrate the anomaly in the respective dataset shifted by the respective overall mean. Small, circle-shaped markers indicate the presence of breakpoints detected by the PMF test only. Breakpoints illustrated by large, stars-shaped markers are also confirmed by the SNH test. A complete list of detected breakpoints is given in Appendix C. Note that some timeseries start between January 1998 and January 2002 or end prior to December 2022, deviating from the GIRAFE v1 coverage.**

In general, an increase in precipitation with an increase in surface temperature is expected, which – in a warming climate – would manifest in a positive trend over time. In particular over the ocean, such an increase is to be expected as the supply of water vapour via evaporation feeding precipitation is not limited. Over land such an increase might be subdued as the supply of water vapour through evapotranspiration and advection is limited (e.g., Roca, 2019). In our analysis, trends are not significant in the majority of cases. In particular, GIRAFE v1 exhibits no significant trend in the considered settings. However,

the considered period is relatively short and, even on these spatially aggregated scales, quite a significant level of variability exists. Both factors impact the significance of trend estimations.

The stability estimates in Table 6 (right block) are all given between GIRAFE v1 and the various listed datasets. GIRAFE v1 is stable with respect to a given dataset if the respective stability value does not deviate significantly from 0 mm/d/decade. In
general, the stability estimates do not exhibit a unique pattern in terms of significance. GIRAFE v1 exhibits negative stabilities over land relative to all other considered datasets. However, the smallest and also non-significant stability occurs relative to GPCC (-0.012 mm/d/decade). Over ocean, where in general a positive trend is expected (see above), GIRAFE v1 shows a non-significant positive trend in contrast to the non-significant negative trend of GPCP v3.2 (left block in Table 6). This discrepancy in the trends explains the significantly non-zero stability estimate (0.048 mm/d/decade) between the two.


**Table 6: Trends (i.e., in timeseries of absolute values in single datasets) in the various datasets (left block) and stabilities of GIRAFE v1 against the respective reference dataset (i.e., trends in timeseries of differences; right block) and their uncertainty in mm/d/decade in the 2002-2020 period. Asterisks (*) indicate significant trends/stabilities.**

| | Trends | | | Stability | | |
|---|---|---|---|---|---|---|
| | ±50°N/S ocean monthly total | ±50°N/S land monthly total | ±30° N/S 99.9 percentile | ±50°N/S ocean monthly total | ±50°N/S land monthly total | ±30° N/S 99.9 percentile |
| GIRAFE v1 | 0.032±0.032 | -0.002±0.025 | -0.89±1.21 | - | - | - |
| GPCP v3.2 | -0.016±0.018 | 0.014±0.020 | -6.96±1.14* | 0.048±0.022* | -0.017±0.016 | 6.07±0.76* |
| ERA5 | -0.009±0.010 | 0.056±0.023* | 6.09±1.02* | 0.041±0.028 | -0.058±0.017* | -6.98±0.86* |
| GPCC v2022 | - | -0.011±0.025 | - | - | -0.012±0.023 | - |
| CMOPRH v1 CRT | -0.009±0.015 | -0.080±0.024* | -1.78±1.11* | 0.041±0.021 | -0.082±0.016* | 0.89±1.01 |
| IMERG v7 FU | 0.044±0.019* | 0.062±0.023* | 0.35±0.94 | -0.011±0.024 | -0.064±0.016* | -1.24±0.66 |

The largest and second largest stability values of GIRAFE v1 for extremes (99.9 percentile, ±30° N/S columns in Table 6) are observed relative to ERA5 and GPCP v3.2, respectively, with opposite signs. We note that GIRAFE v1 does not show confirmed breakpoints (section 5.3) and generally shows a non-significant and smaller trend in the 99.9 percentile (left block in Table 6). The opposite signs in stability of GIRAFE v1 against these two datasets is thereby caused by the decreasing trend in GPCP v3.2 and an increasing trend in ERA5 in extremes as evident in Table 6 (left block) and Figure 10e,f. Again, we can
argue with expectations, i.e., with increasing surface temperatures the extremes in precipitation are expected to increase and not decrease (see also section 5.5). However, results are based on land and ocean observations, with land potentially undergoing an unknown moisture supply. When looking closer at Figure 10e,f, extremes in GPCP v3.2 and ERA5 agree well after 2014 but in the early part of the period both exhibit maximum difference. This approximately coincides with the end of TRMM/TMI

data in 2014/2015. Given that both datasets exhibit largest negative and positive trends, this can be an indication that both are affected by stability issues. At the same time CMOPRH v1 CRT, ERA5 and IMERG v7 FU might also be affected by breakpoints relative to GPCP v3.2 and 3B42 v7 (Figure 10e,f and Appendix C), possibly also caused by the removal of TRMM/TMI in these reference datasets. Sound conclusion on a potential stability issue in GPCP v3.2 and ERA5 are not possible and further analysis is needed. A potential way forward can be to follow approaches outlined in Nguyen et al. (2024), replacing the use of data from neighbouring stations in this study by that of multiple datasets.

We conclude here that GIRAFE v1 has a stability well within the range of available, widely used precipitation datasets, both in terms of monthly total (i.e. mean) and extreme precipitation.

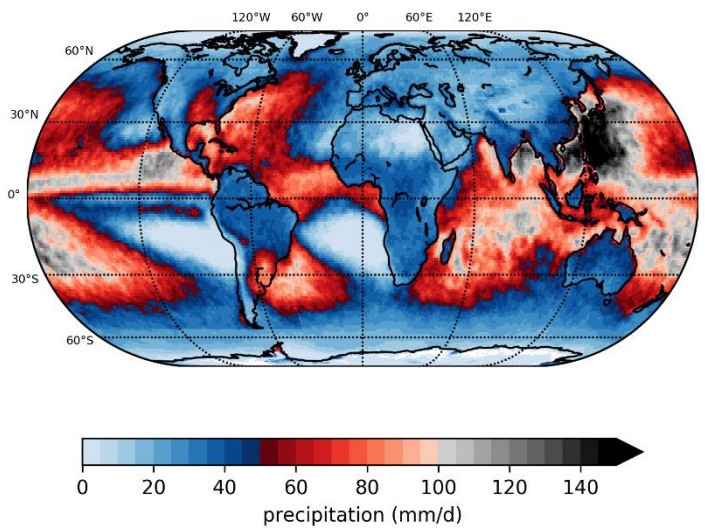

**Figure 11: Map of the 2002-2020 multi-year mean of the annual 99.9th percentile of GIRAFE v1 daily precipitation (mm/d).**

**5.5 Scaling of daily precipitation extremes with surface temperature**

**5.5.1 Qualitative analysis**

Daily extreme precipitation is conventionally defined with the high percentile of the daily wet-days precipitation distribution (Schär et al. 2016). Figure 11 shows the map of the climatology of the annual 99.9th percentile based on GIRAFE v1 data and serves for a discussion of the representation of extreme precipitation in GIRAFE v1 in the following. It is very similar to Rx1day maps (Bador et al., 2020; Alexander et al., 2020), although magnitudes vary across extreme indicators Rx1day and 99.9th percentile as well as across datasets. As expected, it further resembles the climatology of large and organized convective systems (Roca and Fiolleau, 2020), with well characterized maxima in Northern America Great Plains and in South America over Argentina. Orographic features can also be easily spotted. Over ocean, the map recalls the link between intense extremes

and hurricanes as exemplified by the local maximum in the Philippines Sea, Bay of Bengal and Eastern Pacific. The boundary for the in- or exclusion of infrared observations from geostationary satellites at 55°S shows in the Southern Ocean, see also Figure 8 and the related discussions. Regions poleward of 55°N/S, where the absence of geostationary IR observations and the deficiencies of the PMW-derived precipitation rate estimates (cf. section 5.1.2) prevent a correct estimation of the extremes, are likely not well represented in GIRAFE v1, particularly in the southern hemisphere. Beyond this well identified problem, 630 the overall consistency of this diagnostic gives confidence in the ability of GIRAFE v1 to characterize geographical distribution of extreme precipitation.

### 5.5.2 Quantitative analysis

Over the tropical ocean, precipitation is tightly linked to low level moisture. In the case of extreme precipitation, it is possible to further quantify this sensitivity by using SST as a proxy for moisture and simple scaling arguments (Muller and Takayabu 635 2020). Based on these theoretical considerations, an increase of the extreme is expected with SST at a rate of ~6-7%/K following the Clausius-Clapeyron equation. De Meyer and Roca (2021) used the OSTIA SST (Good et al. 2020) and a data pooling technique to estimate this scaling from the observed record. This method has been replicated here. For each SST bin of 0.5 K, all the corresponding wet-days ($P_{acc}$ >= 1 mm/d) precipitation are pooled together. Note that the SST are lagged by 48 h to remove the cooling effect of the precipitation onto the SST (De Meyer and Roca 2021). A five-year period of randomly 640 selected individual, i.e. not necessarily consecutive, years within the 2007-2020 period is used for the data pooling and the 99.9th percentile is estimated from this distribution. Furthermore, a bootstrap of 50 members is constructed using this method. The mean and standard deviation of the 50-member ensemble 99.9th percentile are finally computed.

Figure 12 shows the resulting sensitivity of the 99.9th percentile to the SST for GIRAFE v1 and other products used above for 645 comparison. Between 300.25 K and 302.25 K, where 60-80% of the data points fall, all the products exhibit a linear regime, in agreement with previous studies (De Meyer and Roca, 2021). Over this "Clausius-Clapeyron" regime, extreme precipitation ranges from 120 to 170 mm/d and highlights the difference of magnitude among the products which is smaller than previous assessment based on older products (Roca et al. 2021). GIRAFE v1 stands in the middle, with the CMORPH v1 product being systematically lower and the GPCP v3.2 and IMERG v7 products being larger. As far as the scaling (the slope of the linear 650 regime) is concerned, the products show better agreement than for the ensemble mean estimate of the 99.9th percentile. The scaling is 5.57 ± 0.91 %/K for GPCP v3.2, 5.77 ± 0.93 %/K for GIRAFE v1, 6.38 ± 0.72 %/K for CMORPH v1 and 7.44 ± 0.96%/K for IMERG v7, in good agreement with the Clausius-Clapeyron rate. The small variance of the estimate when bootstrapped over the 14 years period (<1%/K) indicates a weak sensitivity to the selection of the years which in turn imply a weak sensitivity to the configuration of the microwave constellation used in these products in agreement with data-denial 655 experiments (Jucá Oliveira et al. 2022). A discussion of the overall divergence of the datasets at SST < 301 K, in which GIRAFE v1 appears exceptional, can be found in De Meyer and Roca, (2021).

These diagnostics of extreme precipitation consolidate the compliancy of GIRAFE v1 with climate science and monitoring objectives.

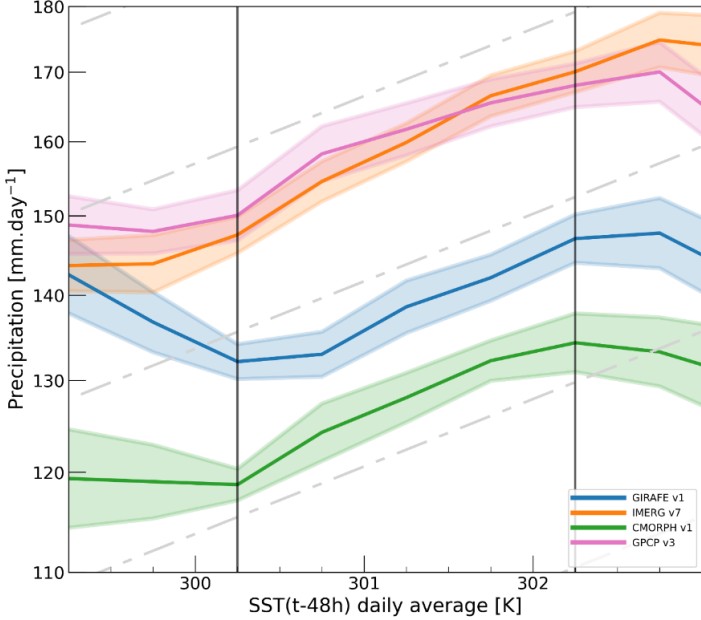

**Figure 12: The 99.9th percentile of the daily distribution as a function of the 48h-lagged underlying SST for several products, including GIRAFE v1. The lines correspond to the 50-member ensemble mean and the shading represent the standard deviation. The vertical lines delineate the central section of the distributions representing the majority of the data (see main text).**

**5.6 Plausibility of the sampling uncertainty**

In this section, the plausibility and usability of the GIRAFE v1 sampling uncertainty is discussed, in terms of i) its agreement with reference datasets within the given uncertainties ("consistency") and ii) its overall distribution and general asymptotic behaviour. The consistency with reference datasets has been briefly covered in the validation against AMMA-CATCH and EURADCLIM (section 5.1). In general, the FEBO of GIRAFE v1 and hence the consistency is larger than that of other 1DD

datasets as these do not feature 1DD uncertainty estimates, hence they can only agree with the reference datasets within the uncertainties of the latter and not the mutual uncertainty as in the case of GIRAFE v1. Figure 13 shows the positive correlation of the GIRAFE v1 sampling uncertainty and the absolute difference in collocated GIRAFE v1 and EURADCLIM 1DD (section 5.1.2). The sampling uncertainty can hence in principle be used as a proxy for the mismatch of GIRAFE v1 from ground-based reference observations. However, the relatively low overall level of the GIRAFE v1 FEBO (60% against AMMA-CATCH

and EURADCLIM in summer months, section 5.1) and the prevalence of GIRAFE v1/EURADCLIM differences exceeding the GIRAFE v1 uncertainty (tilt towards x-axis in Figure 13) indicate that the sampling uncertainty represents less than one sigma on average, i.e., underestimates the full GIRAFE uncertainty. However, systematic shifts in the detection of precipitation and related biases exist in the comparisons against both AMMA-CATCH and EURADCLIM (high FAR coinciding with

positive bias against AMMA-CATCH, section 5.1.1, and low POD coinciding with negative bias against EURADCLIM,
section 5.1.2). The underlying retrieval uncertainties cannot be covered by the GIRAFE v1 sampling uncertainty and explain
at least parts of its above shortcomings.

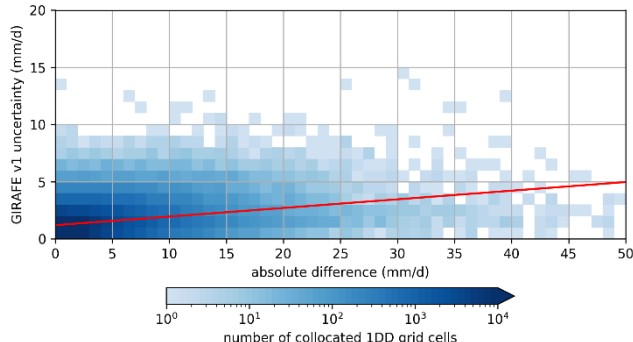

**Figure 13: Two-dimensional distribution (color-coded) of absolute deviations between collocated 1DD GIRAFE v1 and**
**EURADCLIM data (x-axis) and the magnitude of the GIRAFE v1 sampling uncertainty (y-axis). Here, data in 1DD grid cells**
**identified as snow-covered (sections 4.6 and 5.1.2) and with GIRAFE v1 daily precipitation of less than 1 mm have been discarded,**
**i.e., the effect of missed precipitation in GIRAFE v1 (section 5.1.2) is suppressed here. The red line shows the linear least-squares fit**
**to the collocated data points that underlie the distribution, indicating a positive correlation.**

The overall distribution and asymptotic behaviour of the GIRAFE v1 sampling uncertainties are verified against TAPEER
v1.5 sampling uncertainties. The GIRAFE v1 uncertainties are modelled after those of TAPEER, but the underlying PMW
databases differ between GIRAFE v1 and TAPEER v1.5, so the values should not be expected to be identical. Figure 14 shows
the occurrence of 1DD data points along the dimensions of the relative uncertainties in GIRAFE v1 and TAPEER and of daily
precipitation. GIRAFE v1 shares the asymptotic behaviour of TAPEER v1.5 for high 1DD precipitation values. In general,
GIRAFE v1 has narrower distributions along the precipitation dimension (x-axis), especially over ocean. This likely stems
from a general tendency of GIRAFE v1 towards fewer occurrences of high precipitation in GIRAFE v1 as documented by
Konrad et al. (2024). With TAPEER sampling uncertainties extensively studied and validated against ground-based datasets
(Chambon et al., 2013; Gosset et al., 2018), the overall agreement of GIRAFE v1 with TAPEER v1.5 underlines the plausibility
of the GIRAFE v1 sampling uncertainties.

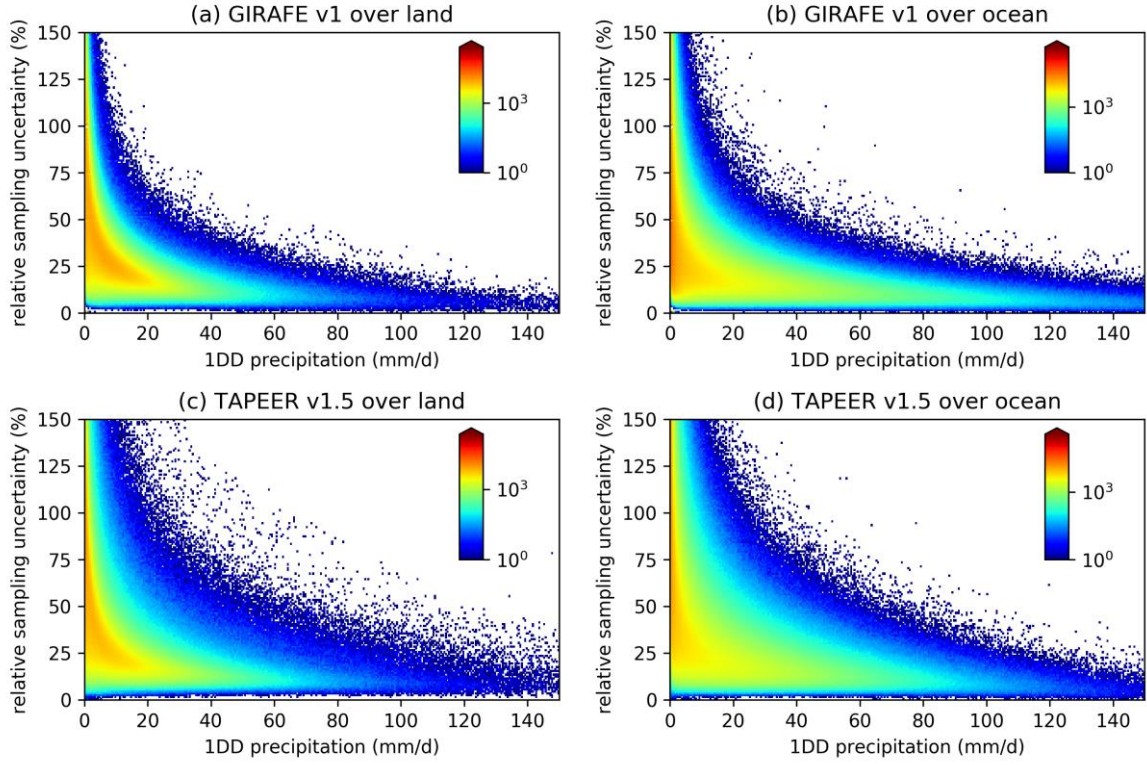

**Figure 14: Two-dimensional histograms of the 1DD relative sampling uncertainties and the 1DD precipitation in GIRAFE v1 (a and b) and TAPEER v1.5 (c and d) from all 1DD grid cells in 2012-2021 inside 30°N/S (TAPEER v1.5 coverage), for which the decorrelation scales had not been replaced by default values in GIRAFE v1 (see section 4.4), separately for land (a and c) and ocean (b and d).**

## 6 Data availability

The data record DOI for GIRAFE v1 is https://doi.org/10.5676/EUM_SAF_CM/GIRAFE/V001 (Niedorf et al., 2024a). Data and associated documentation (scientific references, algorithm theoretical basis documents, validation reports, and user manuals) are available on the above dataset URL.

All intellectual property rights of the CM SAF GIRAFE v1 products belong to EUMETSAT. The use of these products is granted to every interested user, free of charge. If you wish to use these products, EUMETSAT's copyright credit must be shown by displaying the words "copyright (year) EUMETSAT" on each of the products used.

As explained in section 4.3, unofficial variants of the GIRAFE v1 datasets with different start times for one day (6, 12, 18 UTC) exist and can be acquired through the CM SAF user helpdesk at

https://www.cmsaf.eu/EN/Service/UHD/UHD_node.html. Likewise, CM SAF retains the decorrelation scales (section 4.4) as unofficial output for interested expert users.

Feedback on GIRAFE v1 is appreciated by the authors and can be submitted via the CM SAF user helpdesk or the dedicated
forum at https://github.com/cmsaf-girafe.

PMW imager observations are used as input to the HOAPS retrieval (section 3.2.1). For instruments SSM/I and SSMIS the CM SAF SSMI(S) FCDR is available at https://doi.org/10.5676/EUM_SAF_CM/FCDR_MWI/V004. L1C Observations by TMI (V07), AMSR-E (V05) and GMI (V05) are downloaded from https://storm.pps.eosdis.nasa.gov/storm/ and intercalibrated
with the SSMI(S) FCDR as described here: https://www.cmsaf.eu/SharedDocs/Literatur/document/2022/saf_cm_dwd_rep_mii_v1_pdf.html.

PMW sounder observations are used as input to the PNPR-CLIM and PRPS retrievals (sections 3.2.2 and 3.2.1). Observations by instruments AMSU-B and MHS from 2002 to 2017 are available from the FIDUCEO archive at
https://dx.doi.org/10.5285/a8e9f44965434f3b861eba77688701ef. ATMS L1C data and MHS L1C data from 2018 are downloaded from the NASA PPS archive at https://storm.pps.eosdis.nasa.gov/storm/. Precipitation rate obtained from SAPHIR observations by the PRPS retrieval are available at https://dx.doi.org/10.5067/GPM/SAPHIR/MT1/PRPS/2A/06.

ERA5 data as ancillary input to PNPR-CLIM (section 3.2.2) and for constructing the SIF (section 4.6) is available at
https://doi.org/10.24381/cds.f17050d7. Sea and lake ice and SST information from the OSTIA product is available at https://doi.org/10.48670/moi-00168.

MEI v2 data used in the construction of the quantile mapping (section 4.2) have been downloaded from https://psl.noaa.gov/enso/mei/.

The precipitation datasets used for the quality assessment in section 5 are taken from the FROGS database (https://doi.org/10.14768/06337394-73A9-407C-9997-0E380DAC5598) compiled by Roca et al. (2019). Exceptions are EURADCLIM (accessible through https://doi.org/10.21944/1a54-gg96) and AMMA-CATCH (Lebel et al., 2009).

## 7 Conclusions

GIRAFE v1 is a new satellite-based CDR for precipitation, available globally on 1DD and 1DM regular grids from 2002 to 2022. The implementation and generation were carried out by CM SAF in response to the outcome of a dedicated workshop series. The methods for merging PMW and IR observations as presented in section 4.3 are based on TAPEER (Chambon et

al., 2013). The derivation of precipitation rate estimates from PMW observations (section 3.2) is based on various previous works (Andersson et al., 2010; Bagaglini et al., 2021; Kidd et al., 2021).


In terms of product content, GIRAFE v1 stands out with a dedicated and plausible sampling uncertainty, allowing a more robust quantitative analysis of precipitation at 1DD scale. In terms of quality, GIRAFE v1 is able to reproduce reference datasets similar to previously existing products, especially to those that are like GIRAFE v1 not adjusted to ground-based data (section 5.1 and 5.2). Among the (quasi-)global products, GIRAFE v1 stands out in terms of homogeneity (section 5.3) and

stability (section 5.4), at least partially in response to the dedicated post-processing of PMW-based precipitation rate estimates (section 4.2). GIRAFE v1's usefulness for analysing extreme precipitation has been shown in section 5.5. Overall, these analyses underline the compliancy of GIRAFE v1 with requirements for usage in climate monitoring and climate sciences.

The primary uses of GIRAFE v1 include climate monitoring, climate analysis and services, water cycle research, and

evaluating climate models. It is valuable to a wide range of users, including hydro-meteorological services, research institutions, universities, civil and environmental protection agencies, insurance and reinsurance companies, United Nations agencies, water management authorities, agriculture and food production ministries, as well as transportation companies and authorities. These needs align with the WMO Global Framework for Climate Services (GFCS) priority areas, which include agriculture and food security, disaster risk reduction, health, and water availability.


The most apparent limitation of GIRAFE v1 is the low POD of precipitation in mid-to-high latitudes in the presence of surface snow or ice, inherited from the PMW-based precipitation estimates (sections 5.1 and 5.2). Interpretations of GIRAFE v1 data in these situations should be carried out carefully. A respective quality flag allows users to identify potentially affected grid cells. Another limitation is the high fraction of false precipitation during the rainy season in West Africa (section 5.1).


The number of available 1DD grid cells varies over time, due to the constellation of PMW sensors in orbit, with more gaps occurring at the start of the GIRAFE v1 timeseries. The stability of the dataset is affected only to a small degree by the lower amount of PMW-based observations (sections 5.3 and 5.4 and the results by Jucá Oliveira et al. (2022)).

CM SAF provides a framework for the continuous development and operation of CDRs. Hence, the GIRAFE efforts will continue, with an Interim CDR (ICDR) which continues the GIRAFE v1 timeseries from 2023 onwards. CM SAF aims to operationally produce the GIRAFE v1 ICDR in the first half of 2025. Subsequently, efforts will be dedicated to developments towards an improved new version. The latter will address the above-mentioned limitations of GIRAFE v1, the enhancement of uncertainty estimation in addition to the infrared-based sampling uncertainty, a redefinition of the areas in which GIRAFE

works in different ways (currently inside and outside 55°N/S, see section 4.3) and their associated settings (thresholds, extent of local environments), and – using data from qualified sensors – the extension forward and backward in time. Wherever

possible, the prioritisation of improvements will be carried out in close coordination with users during regular workshops. Interested readers are invited to contact CM SAF.

## Appendix A: Effect of quantile mapping

Here, we show the positive effect of the quantile mapping procedure described in section 4.2 on the final GIRAFE output, exemplarily in the case of the precipitation rate estimates by Megha-Tropiques SAPHIR PRPS (MT from here). The original MT distribution sees more rates below 0.5 mm/h than the distribution of the target satellite METOP-A MHS PNPR-CLIM (Figure A1a; blue vs black line). The unadjusted MT observations in the PMW input stream introduce a strong negative bias output compared to GIRAFE output without MT observations ingested (Figure A1b) because the lower ratio of MT 790 observations exceeding the GIRAFE detection threshold of 0.5 mm/h implies fewer precipitation events are detected in the IR observations (section 4.3.1). MT observations are only available from 2012 to 2020, so this situation leads to a temporal instability (discontinuity) when MT enters and leaves the constellation. Quantile mapping brings the MT distribution very close to the target (Figure A1a, red vs black line) from rates of about 0.3 mm/h, i.e. the distributions are aligned at the detection threshold of 0.5 mm/h. The inclusion of the mapped MT data in GIRAFE causes much smaller deviations from the variant not 795 using MT data than without quantile mapping (Figure A1c) and the structural negative bias introduced by MT is mostly remedied. Differences remain which might be linked to deviations in the diurnal sampling between MT and the target satellite or non-representativity of the distributions towards the boundaries of the latitude/longitude bands.

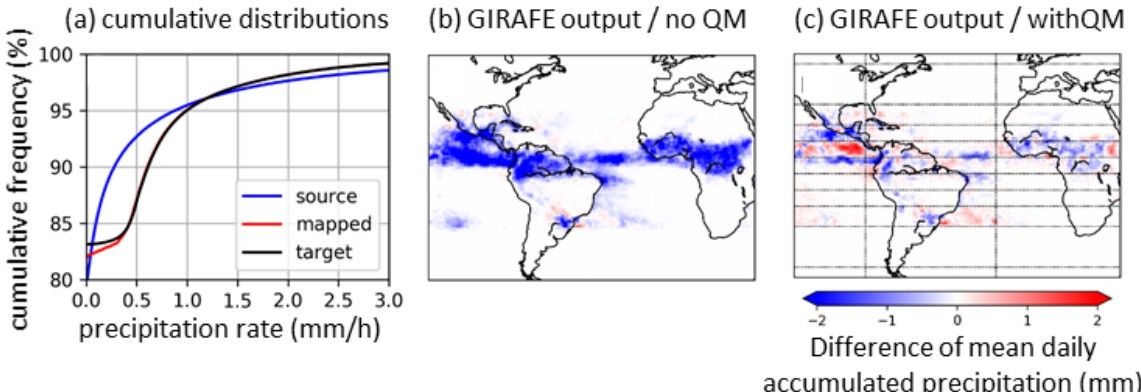

**Figure A1: (a) Exemplary cumulative distributions for METOP-A MHS PNPR-CLIM ("target"; black) and Megha-Tropiques**
**SAPHIR PRPS (MT from here; "source"; blue) from instantaneous observations in June at latitudes inside 4°N/S over South American land. The MT distribution after quantile mapping has been applied ("mapped"; red) is also shown. (b) Differences in GIRAFE daily accumulated precipitation with and without MT instantaneous data averaged over June 2014. Here, quantile mapping has not been applied. (c) Same as (b) but with quantile mapping applied. The lines delineate the latitude/longitude bands in which the cumulative distributions are collected. Note that the results in (b) and (c) depend not only on the data forming the**
**distributions shown in (a), but also on those in the other latitude/longitude bands, depending on the location.**

**Appendix B: Data availability for the comparison against EURADCLIM**

Figure B1 illustrates the mutual availability of the datasets in the comparison against EURADCLIM discussed in section 5.1.2. The comparison of CMORPH v1 RAW against EURADCLIM is based on fewer grid cells because this dataset ends in 2017. The comparison against GIRAFE v1 during summer months and when filtering out SIF-flagged 1DD grid cells is also based on fewer grid cells due to this filtering. Zero availability as illustrated by the grey areas in Figure B1 is mostly caused by the geographical restrictions in the EURADCLIM dataset, except at the 60°N boundary where some of the quasi-global datasets are limited.

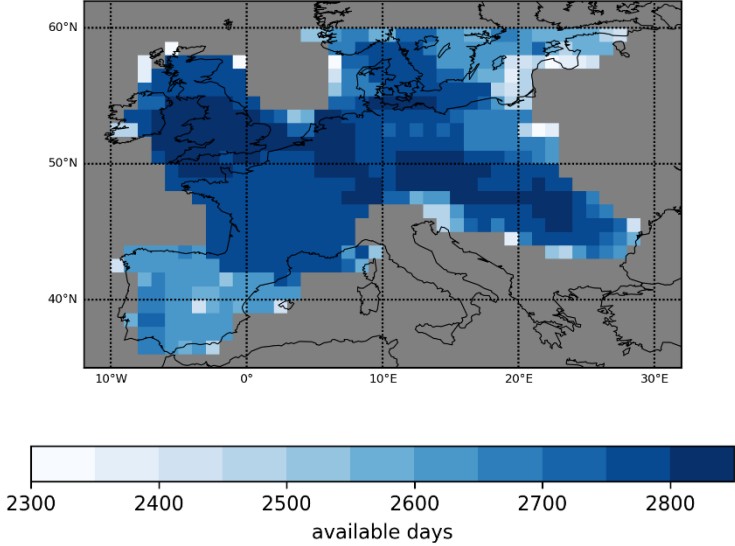

**Figure B1: Number of days during the 2013-2020 period, for which 1DD grid cells are available over Europe in the comparison of 1DD datasets against the ground-based radar dataset EURADCLIM. Grey indicates zero availability.**

**Appendix C: List of breakpoints in the various datasets**

Table C1 lists the breakpoints discussed in sections 5.3 and 5.4 for the respective settings and reference datasets. Note that some breakpoints occur in times preceding the GIRAFE v1 CDR (2002).

**Table C1: Breakpoints at monthly resolution detected by the PMF test as shown as vertical lines in Figure 10 and discussed in sections 5.3 and 5.4; n/a: not available because dataset is respective reference dataset; asterisks (*) indicate break points that are confirmed by the SNH test (bold vertical lines in Figure 10).**

| Dataset | Reference dataset 1 (RD1) | Time of breakpoint w.r.t. RD1 | Reference dataset 2 (RD2) | Time of breakpoint w.r.t. RD2 |
|---|---|---|---|---|
| Monthly totals within 50° N/S over ocean | | | | |

| | | | | |
|---|---|---|---|---|
| GIRAFE v1 | GPCP v3.2 | None | 3B42 v7 | None |
| 3B42 v7 | GPCP v3.2 | 2005-09<br>2007-12<br>2010-08* | 3B42 v7 | n/a |
| CMORPH v1 CRT | GPCP v3.2 | 2007-10<br>2009-10 | 3B42 v7 | 2007-07*<br>2010-09* |
| ERA5 | GPCP v3.2 | None | 3B42 v7 | 2004-08<br>2007-12*<br>2010-09* |
| GPCP v3.2 | GPCP v3.2 | n/a | 3B42 v7 | 2005-09<br>2007-12<br>2010-08* |
| IMERG v7FC | GPCP v3.2 | None | 3B42 v7 | None |
| Monthly totals within 50° N/S over land | | | | |
| GIRAFE v1 | CMORPH v1 CRT | 2020-10 | GPCC FDD v2022 | 2002-10<br>2008-12*<br>2010-06<br>2018-04 |
| 3B42 v7 | CMORPH v1 CRT | 2003-07<br>2004-05*<br>2006-08<br>2008-10*<br>2014-09* | GPCC FDD v2022 | 1998-12<br>2008-10*<br>2010-02<br>2013-11* |
| CMORPH v1 CRT | CMORPH v1 CRT | n/a | GPCC FDD v2022 | 2008-10* |
| ERA5 | CMORPH v1 CRT | None | GPCC FDD v2022 | 1998-12*<br>2008-10*<br>2010-05<br>2020-02 |
| GPCP v3.2 | CMORPH v1 CRT | 2007-10*<br>2020-01 | GPCC FDD v2022 | 2008-10*<br>2010-06*<br>2019-12 |
| IMERG v7FC | CMORPH v1 CRT | None | GPCC FDD v2022 | None |
| GPCC FDD v2022 | CMORPH v1 CRT | 2008-10* | GPCC FDD v2022 | n/a |
| 99.9 percentile within 30°N/S over land and ocean | | | | |
| GIRAFE v1 | GPCP v3.2 | None | 3B42 v7 | None |
| 3B42 v7 | GPCP v3.2 | None | 3B42 v7 | n/a |
| CMORPH v1 CRT | GPCP v3.2 | 2014-05* | 3B42 v7 | 2000-03*<br>2014-09* |
| ERA5 | GPCP v3.2 | 2014-03* | 3B42 v7 | 2014-09* |

| GPCP v3.2 | GPCP v3.2 | n/a | 3B42 v7 | None |
|---|---|---|---|---|
| IMERG v7FC | GPCP v3.2 | 2002-07 2018-10 | 3B42 v7 | 2009-02* 2014-09* |

**Author contributions**

HK prepared the original manuscript with substantial contributions from RR, MS, SF and AN. SF built the archive of geostationary infrared observations. TS and SF implemented the infrared quality control under the guidance of SC. AN, TS, KF and HK built the PMW archive. GP, PS, CK and KF developed the PMW precipitation retrievals and advised on the PMW archive. HK implemented the PMW preprocessing. SF and AN implemented the merging and sampling uncertainty algorithms under the guidance of RR, SC and RAJO. HK, MS, RR, and ML carried out the analyses in section 5. All authors contributed 830 to the revision of the original manuscript.

**Competing interests**

The authors declare that they have no conflict of interests.

**Acknowledgements**

The PNPR-CLIM algorithm has been developed by CNR-ISAC in the C3S_312b_Lot 1 Copernicus Climate Change Service 835 project, for which a licence agreement exists between DWD and CNR-ISAC. This study benefited from the IPSL mesocenter ESPRI facility which is supported by CNRS, UPMC, Labex L-IPSL, CNES and Ecole Polytechnique. The authors thank Leonardo Bagaglini for his contribution in the development of PNPR-CLIM and Adrien Guérou, Marloes Penning de Vries, Ana Radovan, Felix Dietzsch and Marc Pondrom for their contributions to the GIRAFE development at an early stage.

**Financial support**

This work was performed within the EUMETSAT CM SAF framework and the authors acknowledge the financial support of the EUMETSAT member states. The European Union through the ECMWF Copernicus Climate Change Service (C3S) project C3S_312b_Lot1 "Copernicus Climate Change Service: Essential Climate Variable (ECV) products derived from observations Lot 1: precipitation, surface radiation budget, water vapour, cloud properties, and Earth radiation budget" is acknowledged for supporting the development of PNPR-CLIM.

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
