# Peer review of "GIRAFE v1: a global climate data record for precipitation accompanied by a daily sampling uncertainty"

_Earth System Science Data, 2024_

## Referee Comment (RC3)

This study presents the development of the GIRAFE v1 precipitation product, which provides daily accumulations and monthly means at a 1-degree resolution from 2002 to 2022. The product leverages a variety of passive microwave (PMW) radiometers aboard low-Earth-orbit satellites, along with their associated retrieval algorithms, and frequent, high-resolution infrared (IR) observations from geostationary satellites covering all longitudes. The research highlights the importance of GIRAFE v1's homogeneity and stability, as well as its robust capability to quantify sampling uncertainty due to its high temporal resolution at a daily scale. These attributes are significant for climate monitoring and the analysis of climate extremes. However, the manuscript lacks cohesive demonstrations and sufficient rationales throughout its context. Additionally, it would benefit from including more detailed information or figures to enhance comprehensiveness and understanding. Based on the concerns outlined below, I recommend a major revision to address the following comments:

- 1. Line 90: Inconsistent naming between Table 1 and Figure 2 (e.g., MTSAT-1R vs. MTSAT-01R). These should be standardized across all references to ensure consistency and avoid confusion for readers.
- 2. Line 95-100: The phase "Except for.... counts to Tbs" is vague and lacks a clear, explicit explanation. Please provide a more specific elaboration, such as the reason for excluding certain satellites like the Meteosat First Generation.
- 3. Line 115-120: "below 0.3 mm/h are set to zero due to their low signal-to-noise ratio"? Why was 0.3 mm/h selected as the threshold? Is there supporting evidence from a study or reference that validates this choice? A brief explanation of the signal-to-noise ratio in this context would enhance credibility.
- 4. Line 215: "but no automatic detection of such situations has been implemented", This statement suggests manual checking might be used, which is impractical given the vast amount of orbital data. If manual verification is indeed the method, there is a high risk of missing contaminated data periods. Please clarify how data quality is verified or guaranteed under these circumstances. Are there specific quality control measures, such as statistical methods or cross-validation, to minimize errors?
- 5. Line 225: Why PMW observations in the 3° x 3° x 3 days?
- 6. Line 315: "Where the exponential fit procedures for retrieving the decorrelation scales fail, climatological values of 20 km for spatial and 1.5 h for temporal decorrelation are chosen and respective 1DD grid cells are flagged as relying on these default values rather than on the actual variograms." Why? Wha is the rationale?
- 7. Line 360: "For the detection scores, the occurrence of precipitation in a 1DD grid cell is determined at the 1 mm/d threshold." Again, why?

- 8. Section 5.1.1: Validating GIRAFE, a global product, using only one rain gauge network limits the ability to demonstrate its diversity and performance across different regions. Consider including validation data from additional regions or discussing the limitations of this approach
- 9. Section 5.12: For EURADCLIM, adding figures to depict region-wise performance of various products would make the results more intuitive and easier to understand.
- 10. Line 575: Typographical error: "The scaling is  $5.57 \pm 0.91$  %/K for GPCP v3.2 3.2"
- 11. **Comment**: It is important to provide a clear rationale or objective at the beginning of each section before presenting results, rather than starting with phrases like "Here, ..." in Section 5.6 as an example.
- 12. Section 5.6: The description is insufficiently detailed and lacks a strong conclusion. Please elaborate further by providing a deeper analysis of the results.

---

## Author Comment (AC1)

The authors wish to express their sincere thanks to the referees for the efforts they put into the review of our manuscript, which now help us to improve it. We have revised the manuscript in response to the referee comments and are confident that their concerns are sufficiently addressed. We also re-modelled some of the figures for a higher degree of accessibility and consistency. In the following, we respond to the comments directly with our answers to the referee comments in red.

**Anonymous Referee #1**

**GENERAL COMMENTS**

This manuscript reports the creation and analysis of a global rainfall climate data record. By combining observations from LEO PMW and GEO IR, GIRAFE provides the precipitation rate at 1° daily and monthly globally from 2002 to 2022, leveraging three different algorithms (HOAPS, PNPR-CLIM, and PRPS). Uniquely, it includes sampling uncertainties based on decorrelation scales in the IR observations.

This is a generally well-written manuscript introducing a potentially useful dataset for the scientific community. The emphasis on climate-scale consistency is a key advantage of the product and the availability of uncertainty estimates is a unique feature that addresses a longstanding challenge. That being said, I find that the lack of analysis on the uncertainties is an aspect of the manuscript that could be improved. As well, the choice to use PMW observations to determine the precipitation rate and IR observations to calculate the precipitation fraction is somewhat unusual and can benefit from some justification. As such, I recommend Major Revision.

**SPECIFIC COMMENTS**

L85-86: Is there a reason for setting the cutoff at 55°N/S? Geostationary coverage does go further poleward; e.g., the NOAA CPC merged 4-km IR has a 60°N/S coverage while GridSat B1 reaches up to 70°N/S. Also, is the "Geo-Ring" mentioned herein related to the GEO-Ring effort led by EUMETSAT (https://www.eumetsat.int/geo-ring-and-isccp-ng-workshop)? If so, please mention; if not, I suggest renaming the term here to avoid confusion.

With respect to the first part of the comment (why 55°N/S) we have clarified more clearly in section 2 that fields of view are distorted by increasing viewing angles which introduces a trade-off between temporal sampling and accuracy. The boundary at 55°N/S is identical to a similar boundary in GPCP v3.2 for which we added the reference here. The exact value remains somewhat ad hoc, so we also added our plans to redefine the partitions and associated settings in the outlook to future GIRAFE efforts in the very last paragraph of the conclusions section. In the exact instance referenced here (L85-86 in the original manuscript), we have removed the 55° number because the statement is misleading in suggesting that the coverage by geostationary satellites is restricted to 55° latitude. Likewise, we reformulated to not insinuate complete Geo-Ring coverage only from 2002.

With respect to the second part of the comment (the term "Geo-Ring"), GIRAFE utilizes observations from the ring of geostationary satellites over the period 2002-2022. GIRAFE relies on the original L1 data (in terms of calibration and resolution) and applies in-house QC, so at present, the GIRAFE activities are only informally linked with the EUMETSAT activities. GIRAFE might be aligned in the future by using L1 data from the EUMETSAT lead activity. In our opinion, the term Geo-Ring is well suited to describe the constellation of geostationary satellites, but in order to avoid confusion, it has been replaced by "geostationary satellites" or similar in all instances in the manuscript. Section 3.1 clearly states that coverage along all longitudes is achieved for GIRAFE using the described satellites, so the term "ring", implying exactly this complete coverage, can indeed be dropped.

L196: What is the MEI threshold for strong events?

The paragraph has been extended and now gives the thresholds (>1.0 and <1.25) and a caveat that this is an ad hoc definition.

L219-224: This framework of using PMW observations for R and IR observations for F is somewhat counterintuitive compared to the usual method of averaging the precipitation observations from each source. Can the authors discuss their rationale for this framework? This also eases the reader into the following paragraphs, which took me a while to work through because of its unusual approach.

This technique of separating the two estimations of R and F as done here is at the heart of the GOES-Precipitation-Index and has been used for decades. More specifically we follow the Universally Adjusted GPI (UAGPI) framework as described in Xu et al 1999. This choice and the background for such an algorithm is detailed in our core reference (Chambon et al., 2013). Since it is readily available in this reference, we provided only a short summary of the algorithm part. This said, in order to accommodate the remark of the referee we have added a few sentences to provide more context.

L313: Under what circumstance would the exponential fit fail?

This paragraph has been rewritten and explains more clearly what constitutes failure, how this manifests in respective variograms, and how this may relate to the situation (data gaps in the infrared, mostly arid conditions). In response to a comment by Referee #3, we also added the rationale for the use of the "default" values.

L321-322: Using an absolute number of days as an indicator can be problematic since the number of days in each month is not constant; e.g., 10 days means 35.7% of the time in a typical February but 32.2% of the time in January. Why not just provide the number of missing days itself?

The number of available days in a month is indeed provided in the data files, too, which is now clarified in the text. The additional flag allows users to discard monthly mean values according to WMO guidelines which are now referenced in this context in the revised manuscript, too.

Please note that we salso corrected the previously falsely reported number of days at which the flag indicates the violation of WMO guidelines ("more than 10 / more than 4 consecutive").

L477-479: But if the issue is with GPCC, would this not also affect CMORPH and ERA5 (which are independent of GPCC) at the same point as well? And GPCP, which currently has a breakpoint there too, should not have it (since it uses GPCC as an input).

Thank you very much for the careful assessment of this case. The main issue with this figure (and line plots in general) is that the lines are plotted on top of each other, hence become partly invisible in the case of multiple datasets with breakpoints at the same time. In fact, GIRAFE exhibits a break in 2008-12 relative to GPCC over land within 50N/S. ERA5 and CMOPRH exhibit breaks at 2008-10, i.e., within the assumed uncertainty of the method and overlaid by the line for GPCP. Note that GPCC exhibits a break in 2008-10 relative to CMORPH (overlaid by the line for 3B42) but ERA5 does not. This confirms our conclusions that GIRAFE is likely not affected by a break point in 2008-12. However, the GPCP break in 2008-10 remains unclear. We have updated the text and the figure accordingly, the latter now

**showing the breakpoints not as vertical lines but as markers which are set apart from each other along the y coordinate. Appendix C also lists all break points in a separate table.**

Figure 10: I am somewhat surprised that the 99.9th percentile is so low for the Maritime Continent, which appears to be inconsistent with, e.g., the Rx1day figure in Alexander et al. (2020). (But I must admit that it is hard to tell because of the minuscule figure sizes in that paper.) The authors should explain if this is expected behavior.

Alexander al. (2020) show RX1DAY, which is a different marker of extreme compared to the 99.9 utilized here, which may explain discrepancies between figures. This said, overall the tropical land shows differences between the products in the magnitude of the extremes, independently of the markers used. The differences among products is less variable over land than over oceans (see Roca et al., 2025). But still, there are differences in the magnitude of extremes over tropical land between the products, not only over the Maritime Continent. We think that this is out of the scope of the present paper and choose not to expand on these – nevertheless interesting – features including the behavior of the product over the Maritime Continent raised by the referee. Instead we have added a short respective insertion pointing at the occurrence of differences across extreme markers and datasets.

Note that previous Figure 10 is now Figure 11.

Figure 11: While I understand that the main focus is in the middle part of the plot, the left part of the plot (below 300.25 K) caught my eye, since the precipitation-SST relationship for GIRAFE is appreciably different from the other products there. Can the authors explain what may be going on here?

This is discussed in previous papers. De Meyer and Roca (2021) find:

"A "cold" regime with SST < 300 K corresponds to ~ 19 % of the total tropical precipitation amount. In this case, while the results are not sensitive to the SST product selection, or to the timing of the precipitation–temperature association or the length of the record, the various precipitation products exhibit inconsistent behaviors. The lack of robustness of the results might be caused by some structural errors in the precipitation retrievals and/or by the weak data sampling that prevents a robust estimation of the high percentile of the precipitation distribution."

We added a sentence at the end of the paragraph discussing figure 11 (now figure 12) referring the interested reader to this article.

Section 5.6: Given the prominence of uncertainties in this product, I must admit some disappointment in the lack of results on the uncertainties. I was hoping to see a demonstration that the reported uncertainties are indeed representative, for example by performing some regional evaluation and investigating whether the estimates with higher uncertainties have larger errors compared to a reference product. Lacking that, I would like to see some analysis and example use of the uncertainties. If manuscript length is a concern, Sec. 5.4 and 5.5 could be shortened or even removed.

The original manuscript included also the frequency of error bar overlap (FEBO) scores in the validation against AMMA-CATCH and EURADCLIM (section 5.1). Section 5.6 has now been extended to i) discuss these FEBO scores in more detail, and ii) show the correlation between GIRAFE uncertainty and mismatch against EURADCLIM in detail (new figure 13). The rationale around the comparison of overall distributions against TAPEER is extended, too.

In response to a request by Referee #2 we also inserted a reference to use cases for the uncertainty in section 4.4 where the methods for the sampling uncertainty are introduced.

**TECHNICAL CORRECTIONS**

L74-75: I do not see the dashed line in the figure. Also, there is a typo in the figure in the description of "Daily Products".

Thank you very much for spotting this; the reference to the "dashed line" in the caption is a left-over from an earlier version of the figure and has now been removed, as has the typo in the figure.

L88: There is a stray sentence on the loose here.

**Rectified**

L91-92: What do the numbers on the left indicate? The range of longitude of the satellite footprint?

The figure caption has been amended and now clarifies that these number do indeed detail the range of sub-satellite longitudes.

L440-441: The gray lines are practically useless at this point; I can barely see its variation. I suggest removing them, which would allow a reduced y-axis range.

The gray lines are removed as requested and the y-axis range has been narrowed accordingly. The line styles and colors have also been adjusted the sake of accessibility.

Figure 9: In Figure 9c, there is a grayish dotted line in March 2018, which does not appear to correspond to any product in the legend. Please clarify what that is. In general, this diagram is hard to read because of the amount of information and the size of the lines. However, I do not have any ideas what could be done, so I ask the authors to consider if they can improve its legibility.

We have re-done Figure 9 (now Figure 10) and hope that it is more readable now. The specific line in early 2018 mentioned by the referee represents an unconfirmed breakpoint in GIRAFE (small circle-shaped red marker in the new version of the figure). The table in appendix C had been included already in the original manuscript so that breakpoints can be looked up by interested readers where the figure is difficult to read.

L486: The authors should define stability, or at least explain what it means in qualitatively. Otherwise, I do not know how to interpret the numbers in Table 6.

The opening of section 5.4 has been rephrased and now clearly states the definition of stability as the trend in a time series that represents differences between two datasets. The caption of Table 6 is also amended, specifying the difference between "trends" (left block) and "stabilities" (right block).

**Anonymous Referee #2**

**General Comments**

The manuscript presents GIRAFE v1, a newly developed climate data record (CDR) for global precipitation derived from satellite-based infrared and passive microwave observations. The dataset is produced under EUMETSAT's CM SAF initiative and spans the period 2002–2022, with daily and monthly products at 1° spatial resolution. A key advancement is the inclusion of a novel, daily sampling uncertainty estimate—an important addition for climate-oriented applications.

The paper is scientifically robust and technically detailed, offering clear documentation of input data streams, merging methodology, quantile mapping, and validation. The extensive intercomparison with both adjusted and unadjusted satellite precipitation products, as well as the inclusion of quality-focused metrics, adds credibility to the dataset's reliability.

However, a few critical clarifications and improvements are necessary before publication. These mainly concern communication of spatial coverage, formatting issues, and minor technical inconsistencies. I detail these below.

**Specific comments**

The abstract mentions global coverage but omits the crucial latitudinal limit of ±55° for the full IR+PMW merged dataset. Please clarify the spatial domain explicitly for better transparency.

**It is now clarified in the abstract that the combination of IR & PMW is restricted to latitudes inside 55°N/S, and that GIRAFE relies only on PMW observations at higher latitudes.**

While the manuscript provides a detailed derivation of daily sampling uncertainty, it lacks practical interpretation for end users. For example, does high uncertainty imply low confidence or natural variability? Including a brief paragraph on recommended usage or interpretation of this sampling uncertainty would enhance usability.

Good examples of the usage of the sampling uncertainty are found in the case of the comparison between the satellite and the ground-based estimates, both including a sampling uncertainty (Roca et al., 2010; Gosset et al., 2018). Also it has been successfully used to assess the sensitivity of the satellite products to the configuration of the microwave constellation (Roca et al., 2018; Oliveira and Roca, 2022). We have added a paragraph pointing towards these examples at the end of the description of the sampling uncertainty, further extended by mentioning the possibility for usage in discriminating in ensembles of hydrological modelling.

In response to requests by Referees #1 and #3, we also extended section 5.6, e.g. by the analysis of the correlation between GIRAFE mismatch against EURADCLIM and the GIRAFE sampling uncertainty.

In Section 4.2, the authors use quantile mapping to homogenize PMW retrievals across multiple satellite products. It would be helpful to briefly discuss whether this approach fully addresses potential temporal discontinuities in the resulting time series, especially for long-term climate trend analysis.

We clarify in section 4.2 that the drifts of single satellites (or their derived observations) are not corrected by this approach, and that a drift would consequently still lead to discontinuities at either end of the respective satellite's period. We also state that we don't find such discontinuities in our validation.

**Technical Corrections**

There is a typographical error in the flowchart (page 3): 'samling uncertainty' should be corrected to 'sampling uncertainty

**corrected**

L85: The sentence ending with 'partial scans of the Northern and Southern Hemisphere' is followed by a figure caption without sufficient separation. Consider revising the punctuation and improving sentence flow. Also, 'Hemisphere' should be plural in this context.

**corrected**

**Anonymous Referee #3**

This study presents the development of the GIRAFE v1 precipitation product, which provides daily accumulations and monthly means at a 1-degree resolution from 2002 to 2022. The product leverages a variety of passive microwave (PMW) radiometers aboard low-Earth-orbit satellites, along with their associated retrieval algorithms, and frequent, high-resolution infrared (IR) observations from geostationary satellites covering all longitudes. The research highlights the importance of GIRAFE v1's homogeneity and stability, as well as its robust capability to quantify sampling uncertainty due to its high temporal resolution at a daily scale. These attributes are significant for climate monitoring and the analysis of climate extremes. However, the manuscript lacks cohesive demonstrations and sufficient rationales throughout its context. Additionally, it would benefit from including more detailed information or figures to enhance comprehensiveness and understanding. Based on the concerns outlined below, I recommend a major revision to address the following comments:

 Line 90: Inconsistent naming between Table 1 and Figure 2 (e.g., MTSAT-1R vs. MTSAT-01R). These should be standardized across all references to ensure consistency and avoid confusion for readers.

**The naming of satellites in Table 1 and Figure 2 is now unified.**

2. Line 95-100: The phase "Except for.... counts to Tbs" is vague and lacks a clear, explicit explanation. Please provide a more specific elaboration, such as the reason for excluding certain satellites like the Meteosat First Generation.

This paragraph is now rephrase and clarifies more clearly that i) an FCDR archive is of higher quality for climate applications, that ii) GIRAFE – due to its short-window training approach – does not need to have a stable infrared data source, that iii) we have a \*positive\* (and – following the above reasoning – not entirely necessary) exception for MVIRI because there is a published FCDR available, and that iv) we would like to extend the use of FCDRs in future works.

Line 115-120: "below 0.3 mm/h are set to zero due to their low signal-to-noise ratio"? Why was 0.3 mm/h selected as the threshold? Is there supporting evidence from a study or reference that validates this choice? A brief explanation of the signal-to-noise ratio in this context would enhance credibility.

We have extended this statement such that it is clear which processes can affect the detection of precipitation in the sense of a "low signal-to-noise ratio", and refer back to authors of the retrieval algorithm, Andersson et al (2010), for the definition of the 0.3 mm/h threshold.

4. Line 215: "but no automatic detection of such situations has been implemented", This statement suggests manual checking might be used, which is impractical given the vast amount of orbital data. If manual verification is indeed the method, there is a high risk of missing contaminated data periods. Please clarify how data quality is verified or guaranteed under these circumstances. Are there specific quality control measures, such as statistical methods or cross-validation, to minimize errors?

The PMW data are in principle very reliable because they are – over the majority of sensors and times – available as quality-controlled Fundamental Climate Data Records; hence, we have not implemented another – then duplicate – step of quality control. We have clarified that the detection of trends etc that lead to restricting the usage of a satellite in GIRAFE at the start/end of its lifetime is based on non-automated analysis of per-satellite monthly mean anomaly time series.

The specific sentence in the original manuscript that the referee refers to above has been removed because it is inaccurate in suggesting that there are multiple occasions of such short-term data removals when in fact it is only one. We have clarified this and refer to the original documentation where the related problem was reported (Sanó et al (2021)).

We also added a statement that there is no additional layer of quality control on the PMW L1 (Tb) or L2 (precipitation rate estimate) data, and that hence additional shorter periods of contamination might have gone unnoticed.

5. Line 225: Why PMW observations in the 3° x 3° x 3 days?

We extended this statement to explain that this choice together with the choices for rate and detection threshold optimize the frequency distributions of accumulated 1DD precipitation. A similar statement was already present when explaining these threshold values (at the end of sections 4.3.1 and 4.3.2), but lacking the reference to the 3DD environment which is now also included.

6. Line 315: "Where the exponential fit procedures for retrieving the decorrelation scales fail, climatological values of 20 km for spatial and 1.5 h for temporal decorrelation are chosen and respective 1DD grid cells are flagged as relying on these default values rather than on the actual variograms." Why? Wha is the rationale?

In response to a request by Referee #1, the respective paragraph was rewritten and now states more clearly what constitutes a failure. We have also explained that i) the rationale is to avoid gaps, that ii) these "default" values are compatible with the fit results where the fit does not fail, and that iii) the flag allows users to reject respective data points.

7. Line 360: "For the detection scores, the occurrence of precipitation in a 1DD grid cell is determined at the 1 mm/d threshold." Again, why?

1 mm daily accumulation is used as a common threshold for distinguishing between dry and rainy days, also in the context of climate indicators. We added this information about the distinction between "dry" and "rainy" and refer to Gosset et al. (2018).

8. Section 5.1.1: Validating GIRAFE, a global product, using only one rain gauge network limits the ability to demonstrate its diversity and performance across different regions. Consider including validation data from additional regions or discussing the limitations of this approach

We agree with the referee's assessment that the validation against EURADCLIM and AMMA-CATCH cannot necessarily be used for infering the performance of the datasets elsewhere. However, we think that overall global plausibility is sufficiently demonstrated by the intercomparison of (quasi-)global datasets, so that we choose the second option here and add a caveat in section 5.1, prior to sections 5.1.1 and 5.1.2:

"These datasets are limited both in time and more so in space. Already between these two regions, the advantages and disadvantages of the validated datasets vary (see below). It can be expected that regions with other surface settings and climatological conditions will also lead to different results. Hence, the validation here can only be considered a sample rather than complete."

9. Section 5.12: For EURADCLIM, adding figures to depict region-wise performance of various products would make the results more intuitive and easier to understand.

We have added one exemplary figure for bias corrected RMSD for the various datasets (new figure 7) and discuss respective results in section 5.1.2.

10. Line 575: Typographical error: "The scaling is  $5.57 \pm 0.91$  %/K for GPCP v3.2 3.2"

**corrected**

11. Comment: It is important to provide a clear rationale or objective at the beginning of each section before presenting results, rather than starting with phrases like "Here, ..." in Section 5.6 as an example.

We agree that sections should be properly introduced and have re-phrased the beginnings of several sections (4, 5.1.1, 5.3, 5.4, 5.5.1, 5.6).

12. Section 5.6: The description is insufficiently detailed and lacks a strong conclusion. Please elaborate further by providing a deeper analysis of the results.

We agree that section 5.6 was weak and too short. We extended it by a proper introduction (see also our response to the previous comment by the referee), additional analysis (also in response to requests by Referees #1 and #2), and clearer statements on the outcome of the analyses.

**References by the authors**

Alexander, L. V., Bador, M. Roca, R., Contractor, S. Donat, M.G. and Nguyen, P. L.: Intercomparison of Annual Precipitation Indices and Extremes over Global Land Areas from in Situ, Space-Based and Reanalysis Products. Environ. Res. Lett. 15 (5). https://doi.org/10.1088/1748-9326/ab79e2, 2020.

Andersson, A., Fennig, K., Klepp, C., Bakan, S., Graßl, H., and Schulz, J.: The Hamburg Ocean Atmosphere Parameters and Fluxes from Satellite Data – HOAPS-3, Earth Syst. Sci. Data, 2, 215–234, https://doi.org/10.5194/essd-2-215-2010, 2010.

Chambon, P., Jobard, I., Roca, R., and Viltard, N.: An investigation of the error budget of tropical rainfall accumulation derived from merged passive microwave and infrared satellite measurements, Q. J. Roy. Meteor. Soc., 139, 879–893, https://doi.org/10.1002/qj.1907, 2013.

Gosset, M., Alcoba, M., Roca, R., Cloché, S., and Urbani, G.: Evaluation of TAPEER daily estimates and other GPM-era products against dense gauge networks in West Africa, analysing ground reference uncertainty. Q. J. R. Meteorol. Soc., 144: 255–269. https://doi.org/10.1002/qj.3335, 2018.

Oliveira, R. A. J., and Roca, R: A Simple Statistical Model of the Uncertainty Distribution for Daily Gridded Precipitation Multi-Platform Satellite Products. Remote Sens. 14 (15). https://doi.org/10.3390/rs14153726, 2022.

Roca, R., Chambon, P., Jobard, I., Kirstetter, P., Gosset, M. and Bergès, J. C.: Comparing Satellite and Surface Rainfall Products over West Africa at Meteorologically Relevant Scales during the AMMA Campaign Using Error Estimates, J. Appl. Meteor. Climatol., 49, 715–731, https://doi.org/10.1175/2009JAMC2318.1, 2010.

Roca, R., Taburet, N., Lorant, E., Chambon, P., Alcoba, M., Brogniez, H., Cloché, S., Dufour, C., Gosset, M., and Guilloteau, C.: Quantifying the contribution of the Megha-Tropiques mission to the estimation of daily accumulated rainfall in the Tropics, Q. J. R. Meteorol. Soc., 144, 49–63, https://doi.org/10.1002/qj.3327, 2018.

Roca et al., 2025: GEWEX/IPWG Precipitation Assessment: An Update on Extreme Daily Precipitation over Tropical Ocean, GEWEX Quaterly, Vol. 34, No. 4 | Quarter 4 2024)

Sanò, P. Panegrossi, G., Bagaglini, L., Cattani, E., Konrad, H., Sikorski, T., Schröder, M.: COBRA: Algorithm Theoretical Basis Document, https://confluence.ecmwf.int/pages/viewpage.action?pageId=278552349, 2021.

Xu L, Gao X, Sorooshian S, Arkin PA, Imam B. 1999. A microwave infrared threshold technique to improve the GOES precipitation index. J. Appl. Meteorol. 38: 569–579.